# Replica Bethe Ansatz solution
# to the Kardar-Parisi-Zhang equation on the half-line

**Alexandre Krajenbrink[1,2⋆] and Pierre Le Doussal[2]**

**1** SISSA and INFN, via Bonomea 265, 34136 Trieste, Italy
**2** Laboratoire de Physique de l'École Normale Supérieure, ENS,
Université PSL, CNRS, Sorbonne Université, Université Paris-Diderot,
Sorbonne Paris Cité, 24 rue Lhomond, 75005 Paris, France

⋆ krajenbrink@ens.fr

## Abstract

We consider the Kardar-Parisi-Zhang (KPZ) equation for the stochastic growth of an interface of height $h(x,t)$ on the positive half line with boundary condition $\partial_x h(x,t)|_{x=0} = A$. It is equivalent to a continuum directed polymer (DP) in a random potential in half-space with a wall at $x = 0$ either repulsive $A > 0$, or attractive $A < 0$. We provide an exact solution, using replica Bethe ansatz methods, to two problems which were recently proved to be equivalent [Parekh, arXiv:1901.09449]: the droplet initial condition for arbitrary $A \geqslant -1/2$, and the Brownian initial condition with a drift for $A = +\infty$ (infinite hard wall). We study the height at $x = 0$ and obtain (i) at all time the Laplace transform of the distribution of its exponential (ii) at infinite time, its exact probability distribution function (PDF). These are expressed in two equivalent forms, either as a Fredholm Pfaffian with a matrix valued kernel, or as a Fredholm determinant with a scalar kernel. For droplet initial conditions and $A > -\frac{1}{2}$ the large time PDF is the GSE Tracy-Widom distribution. For $A = \frac{1}{2}$, the critical point at which the DP binds to the wall, we obtain the GOE Tracy-Widom distribution. In the critical region, $A + \frac{1}{2} = \epsilon t^{-1/3} \to 0$ with fixed $\epsilon = \mathcal{O}(1)$, we obtain a transition kernel continuously depending on $\epsilon$. Our work extends the results obtained previously for $A = +\infty$, $A = 0$ and $A = -\frac{1}{2}$.

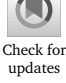

# 1 Introduction

The continuum Kardar-Parisi-Zhang (KPZ) equation in one dimension [1–7] describes the stochastic growth of an interface parameterized by a height field $h(x,t)$ at point $x \in \mathbb{R}$ and time $t$. In appropriate units it reads

$$\partial_t h(x,t) = \partial_x^2 h(x,t) + (\partial_x h(x,t))^2 + \sqrt{2}\,\xi(x,t)\,, \tag{1}$$

where $\xi(x,t)$ is a centered Gaussian white noise with $\mathbb{E}\big[\xi(x,t)\xi(x',t')\big] = \delta(x-x')\delta(t-t')$. An important question is to determine the probability distribution function (PDF) for the height at one point, $h(0,t)$, given an initial condition $h(x,t=0)$. Exact solutions, valid for all times $t > 0$, have been obtained for several initial conditions, notably droplet, flat and Brownian (including stationary) [8–16]. Remarkably they are expressed using Fredholm determinants

or Pfaffians. From them it was shown that, in the large time limit, the PDF's of the shifted height fluctuations, $H(t) = h(0, t) + \frac{t}{12} \sim t^{1/3}$, are described by the Tracy Widom distributions [17, 18], i.e. the distributions of the largest eigenvalues of standard Gaussian random matrix ensembles [19, 20]. Some of these results were obtained using the replica Bethe ansatz (RBA) method. This method, pioneered by Kardar [21], consists of several steps. First one maps, via the Cole-Hopf transformation $Z(x, t) = e^{h(x,t)}$, the KPZ equation for $h(x, t)$ to the stochastic heat equation (SHE) for $Z(x, t)$, which can then be interpreted as the partition function of a continuum directed polymer of length $t$ in a random potential with final point at $x$. Next one expresses the $n$-th integer moment of $Z(x, t)$ using (i) a mapping to the attractive delta Bose gas with $n$ bosons (the replica) in one dimension, and (ii) its exact solution from the Bethe ansatz. From the moments one then obtains the generating function (i.e. the Laplace transform of the PDF of $Z(x, t)$), and at large time, the PDF of $h(x, t)$.

Here, we consider the KPZ equation on the half-line, where Eq. (1) is considered for $x \in \mathbb{R}^+$ along with the Neumann boundary condition (b.c.)

$$\forall t > 0, \quad \partial_x h(x, t)|_{x=0} = A \quad \Longleftrightarrow \quad (\partial_x - A)Z(x, t)|_{x=0} = 0, \tag{2}$$

where $A$ is a real parameter which describes the interaction with the boundary (a wall at $x = 0$). This problem was considered in a pioneering paper by Kardar [22] in the equivalent representation in terms of a directed polymer in a half space bounded by a wall. The wall is repulsive for $A > 0$ and attractive for $A < 0$. The case $A = +\infty$ imposes $Z(x = 0, t) = 0$, an infinitely repulsive wall. It can also be seen as absorbing wall, while $A = 0$ can be seen as a reflecting wall. A binding transition to the wall was predicted for $A = -1/2$ as $A$ is decreased, from heuristic considerations on the ground state of the associated delta Bose gas in presence of a wall. It was later observed in numerical simulations of a discretized model [23]. More recently, exact results for three specific values of $A$, i.e. $A = +\infty, 0, -1/2$, were obtained [24–26] for the droplet IC defined as

$$h(x, t = 0) = -\frac{|x - \kappa|}{\eta} - \log \eta \quad, \quad \eta \to 0^+ \quad \Longleftrightarrow \quad Z(x, t = 0) = \delta(x - \kappa). \tag{3}$$

What was studied is the height at the origin, $H(t)$, i.e. $H(t) = h(\kappa = 0, t)$ for finite $A$, and $H(t) = h(\kappa, t) - \log \kappa$ with $\kappa \to 0^+$ since a regularization is needed for $A = +\infty$ (see below). In all three cases the solution can be expressed in terms of a Fredholm Pfaffian involving a matrix kernel [24–27]. For $A = +\infty$, the infinitely repulsive wall, it was found [24] that the PDF of the scaled height, $H(t)/t^{1/3}$, converges at large $t$ to the Tracy-Widom distribution associated to the Gaussian Symplectic Ensemble (GSE) of random matrices [18–20]. Note that the original RBA solution of [24] involves a scalar kernel, but recently an equivalent solution was obtained by us in terms of a matrix kernel [27]. For $A = 0$, although the finite time matrix kernel differs from the case $A = +\infty$, it was found that the large time limit of the PDF also corresponds to the TW-GSE distribution [25]. Both cases used the mapping to the delta Bose gas, with use of respectively the RBA for $A = +\infty$ and nested contour integral representations of the moments for $A = 0$. The case $A = -\frac{1}{2}$, i.e. the critical case, was solved instead using a continuum limit from the ASEP model with an open boundary [26]. It was found that at large time the PDF converges to the Tracy Widom distribution associated to the Gaussian Orthogonal Ensemble (GOE) which thus describes the large time behavior at the transition.

Until now the KPZ equation for general $A$ has resisted the analysis. The RBA approach of [24] has been extended using that the associated Bose gas is integrable for general $A$. This led to an explicit formula for the $n$-th integer moment of $Z$ [28], which will be fully confirmed here, but valid however only in some restricted range of $n$ and $A$, namely $A + 1/2 > n/2$. The origin of this restriction is that for any finite $A$, a complicated structure of multi particle bound

states to the wall arises in the spectrum of the associated delta Bose gas. This problem did not impact the study in [24] since for $A = +\infty$ there are no boundary bound states. Extracting the PDF from the moments in the restricted range has eluded previous attempts, but here we will provide the solution to this puzzle. Another approach was followed recently: the full structure of these bound states was obtained [29] leading to an improved moment formula a priori valid for all $A, n$. A (quite formal) Pfaffian formula was obtained from these moments. Among the partial results obtained, the large time limit as a GSE-TW distribution for all $A > -1/2$, and the convergence to a Gaussian PDF in the bound phase $A < -1/2$ [29]. This study however failed until now to produce the full finite time answer, and also to obtain the large time behavior in the critical region.

The half-space problem has also been addressed for other solvable models in the KPZ universality class, mostly in the mathematics literature. In a pioneering paper, Baik and Rains [30] studied the longest increasing sub-sequences (LIS) of *symmetrized* random permutations. The problem maps to a discrete zero temperature model of a directed polymer in a half-space, with a tunable parameter $\alpha$ which makes the boundary more attractive as $\alpha$ increases. They found, in the limit of large polymer length $t$, a transition when $\alpha$ reaches the critical value $\alpha_c = 1$. For $\alpha < \alpha_c$ the PDF of the fluctuations of the DP energy (analog to the height in a growth model) is given by the Tracy-Widom GSE distribution [18] on the characteristic KPZ scale $t^{1/3}$. For $\alpha > \alpha_c$ the PDF is Gaussian on the scale $t^{1/2}$, as the DP paths are bound to the diagonal line. At the critical point, $\alpha = \alpha_c$ the PDF is given by the GOE Tracy-Widom distribution on the $t^{1/3}$ scale. A similar transition was found for the height distribution in the discrete PNG growth model on a half-line, with a source at the origin, as the the nucleation rate at the origin is increased above a threshold [31, 32]. Results were also obtained for the TASEP in a half-space [33]. Finally, formula were obtained for the (finite temperature) log-gamma DP with symmetric weights [34, 41] and half-quadrant geometries [35] with, however, not yet rigorous asymptotic results. In the case of the ASEP model in a half space a Bethe ansatz study was carried in Ref. [36] without however asymptotic results. In the case $A = -1/2$, the ASEP and its KPZ limit were studied using half-space Macdonald processes [41] and TW GOE asymptotics were proved, see Refs. [26, 58].

It is natural to conjecture that the transition for the KPZ equation at $A = -1/2$ is in the same universality class, in the large time limit, as the one discovered by Baik and Rains in [30] and that this universality is common to the full KPZ class, see Ref. [58]. Baik and Rains performed a detailed analysis on a scale $\alpha - 1 = w t^{-1/3}$ around the transition. They found that the PDF depends continuously on $w$ and interpolates between the GSE/GOE/Gaussian distributions as $w$ is increased. This transition PDF was obtained as a solution of Painlevé type system of equations. Further results were obtained recently using Pfaffian-Schur processes, for variants of TASEP models and last passage percolation in a half-quadrant [37, 38]. Not only the one-point, but also the multi-point height distributions where studied (the extended process) and for arbitrary positions with respect to the wall. A Fredholm Pfaffian formula was obtained with an explicit expression for the (extended) matrix kernel around the GSE/GOE/Gaussian transition. One may conjecture that it is compatible with the Painlevé system of [30] but this has not yet been verified. Finally numerical studies have addressed the half-line problem. In [39] the convergence to the GSE was explored in a half-space geometry aimed to open the way for an experimental confirmation. In [23] the Baik Rains transitional PDF was verified and connections to conductance fluctuations in Anderson localization were explored.

The aim of this paper is to provide an exact solution for the KPZ equation on a half-line for generic values of $A$ using the replica Bethe ansatz. We study the height at $x = 0$ and obtain (i) at all time the generating function (i.e. the Laplace transform of the PDF of $Z = e^h$) (ii) at infinite time, its exact probability distribution function (PDF). These are expressed in two

equivalent forms, either as a Fredholm Pfaffian with a matrix valued kernel, or as a Fredholm determinant with a scalar kernel. For droplet initial conditions and $A > -\frac{1}{2}$ the large time PDF is the GSE Tracy-Widom distribution. We obtain an explicit formula for the transition kernel in the critical region, $A + \frac{1}{2} = \epsilon t^{-1/3} \to 0$ with fixed $\epsilon = \mathcal{O}(1)$, continuously depending on $\epsilon$. Although it is different from the expression obtained in [37, 38], we conjecture that the two expressions are in fact equivalent, and check it by a systematic expansion in traces to third order. In the limit $\epsilon \to 0$ we obtain that the PDF converges to the GOE TW distribution. Our results thus show universality and that the KPZ equation in the half-line is in the KPZ class. Note however that for a technical reason, we are able to present the result only for the case $A \geqslant -1/2$, i.e. $\epsilon \geqslant 0$.

Our strategy is the following. It is inspired by a theorem shown recently:

**Theorem 1.1** (Parekh. Theorem 1.1 from [40]). *Let $Z(0, x)$ denote the solution of the Stochastic Heat Equation (SHE) with Robin-boundary parameter $A \in \mathbb{R}$ and droplet initial condition $Z(x, 0) = \delta(x)$. Let $Z_{\mathrm{Br}}(x, t)$ the solution to the SHE with Brownian initial condition with drift $A + \frac{1}{2}$ i.e. $Z_{\mathrm{Br}}(x, 0) = e^{B(x) - (A + \frac{1}{2})x}$, where $B$ is a (zero drift) Brownian motion, with Dirichlet boundary condition $Z_{\mathrm{Br}}(0, t) = 0$. Then we have the following equality in distributions*

$$Z(0, t) = \lim_{\kappa \to 0} \frac{Z_{\mathrm{Br}}(\kappa, t)}{\kappa}. \tag{4}$$

*Remark* 1.2. This theorem comes as a limit of an identity proved in Ref. [41], Proposition 8.1, on half-space Macdonald processes.

We thus choose to study, rather than our original problem with parameter $A$, the KPZ problem on the half-line with Dirichlet boundary conditions, but with Brownian initial conditions. To be able to apply the theorem we choose the drift of the Brownian to be $A + \frac{1}{2}$. Since the boundary conditions are Dirichlet (i.e. $A = +\infty$) we can apply the same RBA method as in [24] (there are no boundary bound states for $A = +\infty$). The only technical difficulty is the calculation of the "overlap" of the Brownian initial condition with the eigenstates of the delta Bose gas, which we are able to perform. We then obtain a formula for the $n$-th integer moment of $Z_{\mathrm{Br}}(x, 0)$, which then leads us to a formula for the moments of the droplet initial condition for generic $A$ using the theorem. Although obtained via a completely different calculation the formula is, in the end, identical to the one obtained in [28] (which, in a sense, confirms the theorem). It is also valid only in the same restricted range $A + \frac{1}{2} > n/2$ (as the overlap diverges beyond). This is a well known difficulty, which arises already on the full line, but was surmounted in Refs [14, 15] to obtain the solution of KPZ with Brownian initial conditions on the full line. We thus follow the same strategy, and it leads us to the abovementioned results.

Let us close by mentioning that these exact formulae for the KPZ equation at all times are very useful to obtain exact results for the large deviations of the PDF of the KPZ height field both at large time [46–49] and short time [50–53] and in particular in the half-space [27] with excellent agreeement with numerics [54, 55]. These exact solutions have also been used in the mathematics community to prove exact bounds on the tails of the PDF of the KPZ height, see Refs. [42–45]. Although we will study here only typical fluctuations and not large deviations, the new formulae obtained in this work should also allow for such results for generic $A$ in the future.

## 2  Presentation of the main results

In this paper we obtain in a single unified calculation the statistics of

(i) $Z(t) = e^{h(t)}$ and $H(t) = h(0,t) + \frac{t}{12}$ where $h(0,t)$ is the height at $x = 0$ with droplet initial condition in presence of a wall of parameter $A$ ;

(ii) $\hat{Z}_{\mathrm{Br}}(t) = e^{h_{\mathrm{Br}}(t)} = \lim_{\kappa \to 0^+} e^{h_{\mathrm{Br}}(\kappa,t)}$ and $H_{\mathrm{Br}}(t) = h_{\mathrm{Br}}(\kappa,t) + \frac{t}{12}$ where $h_{\mathrm{Br}}(\kappa,t) - \log \kappa$ is the height at $x = \kappa$ with Brownian initial condition with a drift $A + \frac{1}{2}$ in presence of a hard-wall.

From the theorem of Parekh in [40] one has the equality of the two generating functions, for any $\varsigma > 0$

$$\mathbb{E}_{\mathrm{KPZ,Brownian}}\left[\exp(-\varsigma e^{H_{\mathrm{Br}}(t)})\right] = \mathbb{E}_{\mathrm{KPZ}}\left[\exp(-\varsigma e^{H(t)})\right]. \tag{5}$$

The expected value of the left hand side is taken over the white noise of the KPZ equation and the Brownian initial condition while the expected value of the right hand side is taken over the white noise of the KPZ equation.

## 2.1 Finite time

Our main result valid for all time $t \geqslant 0$ and all $A > -\frac{1}{2}$ is that the generating function for $\varsigma > 0$ can be written as a Fredholm Pfaffian

$$\mathbb{E}_{\mathrm{KPZ}}\left[\exp(-\varsigma e^{H(t)})\right] = 1 + \sum_{n_s=1}^{+\infty} \frac{(-1)^{n_s}}{n_s!} \prod_{p=1}^{n_s} \int_{\mathbb{R}} \mathrm{d}r_p \frac{\varsigma}{\varsigma + e^{-r_p}} \mathrm{Pf}\left[K(r_i, r_j)\right]_{n_s \times n_s}, \tag{6}$$

where kernel $K$ is matrix valued and represented by a $2 \times 2$ block matrix with elements

$$K_{11}(r,r') = \iint_{C^2} \frac{\mathrm{d}w}{2i\pi} \frac{\mathrm{d}z}{2i\pi} \frac{w-z}{w+z} \frac{\Gamma(A+\frac{1}{2}-w)}{\Gamma(A+\frac{1}{2}+w)} \frac{\Gamma(A+\frac{1}{2}-z)}{\Gamma(A+\frac{1}{2}+z)}$$
$$\times \Gamma(2w)\Gamma(2z)\cos(\pi w)\cos(\pi z)e^{-rw-r'z+t\frac{w^3+z^3}{3}},$$

$$K_{22}(r,r') = \iint_{C^2} \frac{\mathrm{d}w}{2i\pi} \frac{\mathrm{d}z}{2i\pi} \frac{w-z}{w+z} \frac{\Gamma(A+\frac{1}{2}-w)}{\Gamma(A+\frac{1}{2}+w)} \frac{\Gamma(A+\frac{1}{2}-z)}{\Gamma(A+\frac{1}{2}+z)}$$
$$\times \Gamma(2w)\Gamma(2z)\frac{\sin(\pi w)}{\pi}\frac{\sin(\pi z)}{\pi}e^{-rw-r'z+t\frac{w^3+z^3}{3}}, \tag{7}$$

$$K_{12}(r,r') = \iint_{C^2} \frac{\mathrm{d}w}{2i\pi} \frac{\mathrm{d}z}{2i\pi} \frac{w-z}{w+z} \frac{\Gamma(A+\frac{1}{2}-w)}{\Gamma(A+\frac{1}{2}+w)} \frac{\Gamma(A+\frac{1}{2}-z)}{\Gamma(A+\frac{1}{2}+z)}$$
$$\times \Gamma(2w)\Gamma(2z)\cos(\pi w)\frac{\sin(\pi z)}{\pi}e^{-rw-r'z+t\frac{w^3+z^3}{3}},$$

$$K_{21}(r,r') = -K_{12}(r',r).$$

In this formula the contours $C$ are parallel to the imaginary axis and cross the real axis between $0$ and $A + \frac{1}{2}$. For definition and more details about Fredholm Pfaffians see the end of Section 3.3. For a visual illustration of the block structure in the Pfaffians of matrix valued kernel, appearing in (6), see e.g. (146).

*Remark* 2.1. This formula reproduces the known cases. For $A \to +\infty$, the ratio of $\Gamma$ functions containing $A$ behaves as $\sim e^{-2(w+z)\log A}$. A change of variable $r = \tilde{r} - 2\log A$ then recovers the formula that we obtained in Eq. (5) and (6) of [27] with the correspondence $H_1 = H(t) + 2\log A$ (up to a trivial rescaling involving time explained below Eq. (62) in [27]). There it was shown that this formula is equivalent to the result obtained in [24] in terms of a scalar kernel. Our result for $A = 0$, using some elementary manipulations [1] identifies with the one in [25]. The limit $A = -\frac{1}{2}$ of our formula is more involved, and we discuss it below.

---

[1]We first rewrite (7) using $\frac{\sin \pi z}{\pi} = \frac{1}{\Gamma(z)\Gamma(1-z)}$, $\cos \pi z = \frac{\pi}{\Gamma(\frac{1}{2}+z)\Gamma(\frac{1}{2}-z)}$, $\Gamma(2z) = \Gamma(z)\Gamma(z+\frac{1}{2})2^{2z-1}/\sqrt{\pi}$ for $z$ and for $w$. We can now compare with formula (5) and below in [25] (arXiv version). One finds $K_{22}(r,r') = \tilde{K}_{11}(-r,-r')/\pi$ (using the change $(w_1, w_2) \to (-w, -z)$) $K_{11}(r,r') = \tilde{K}_{22}(-r,-r')\pi$ (using the change $(s_1, s_2) \to (w,z)$) and

The series in Eq. (6) can also be interpreted as a Fredholm Pfaffian, see Eq. (54), implying a duality between the generating function of the exponential KPZ height and an average of a "Fermi factor" over a Pfaffian point process (see Ref. [27] Eq. 7 for instance for further details on this type of duality)

$$\mathbb{E}_{\text{KPZ}}\Big[\exp(-\varsigma e^{H(t)})\Big] = \mathbb{E}_K\left[\prod_{i=1}^{+\infty}\frac{1}{1+\varsigma e^{a_i}}\right],\tag{8}$$

where the set $\{a_i\}$ forms a Pfaffian point process with kernel $K$.

## 2.2 Large time limit

At large time, we have obtained the following limit behavior of the solution of the Kardar-Parisi-Zhang equation:

- For any $A > -\frac{1}{2}$, the one-point KPZ height fluctuations follow the Tracy-Widom GSE distribution

$$\lim_{t\to\infty}\mathbb{P}\big(\frac{h(0,t)+\frac{t}{12}}{t^{1/3}}\leqslant s\big) = F_4(s).\tag{9}$$

- For $A = -\frac{1}{2}$, the one-point KPZ height fluctuations follow the Tracy-Widom GOE distribution.

$$\lim_{t\to\infty}\mathbb{P}\big(\frac{h(0,t)+\frac{t}{12}}{t^{1/3}}\leqslant s\big) = F_1(s).\tag{10}$$

*Remark* 2.2. Here $F_4(s) = \frac{1}{2}(F_1(s) + \frac{F_2(s)}{F_1(s)})$ is the cumulative distribution function (CDF) of the GSE-TW distribution, as defined in [56]. Another convention, which we denote $\tilde{F}_4$ with $F_4(s) = \tilde{F}_4(\frac{s}{\sqrt{2}})$, is given in [18, 57].

Near the transition point $A = -1/2$, there is critical regime for $A + \frac{1}{2} \to 0$ and $t \to +\infty$ simultaneously, with the crossover parameter $\epsilon = (A+\frac{1}{2})t^{1/3}$ being kept fixed and $\mathcal{O}(1)$. From the above formula (7) taking $\varsigma = e^{-st^{1/3}}$ one obtains the large time limit of the matrix kernel. This limit kernel, which depends continuously on $\epsilon$, is called the transition matrix kernel. Its expression, $K^\epsilon$, is given in two equivalent forms in Eqs. (64) and in Eqs. (71) for $\epsilon > 0$. In the limit $\epsilon \to +\infty$ this transition kernel becomes equal to the standard matrix kernel of the GSE (the GSE limit is also obtained by taking the large time limit for any fixed $A > -1/2$ directly from (7)). The transition at $A = -1/2$ for the KPZ equation is believed to be in the same universality class than the one for last passage percolation in a half-quadrant. For the latter an explicit Fredholm Pfaffian formula was obtained in Ref. [37,38]. Although the kernel there is different from ours, we have checked by expansion in traces to third order that for $\epsilon > 0$ their Fredholm Pfaffian coincides (see Appendix C).

In Ref. [27] we presented a general procedure to construct a scalar kernel from a matrix valued kernel with a Schur Pfaffian structure. Here we use this method to obtain the CDF of the one-point KPZ height in terms of a Fredholm determinant involving the following scalar kernel

$$\lim_{t\to\infty}\mathbb{P}\big(\frac{H(t)}{t^{1/3}}\leqslant s\big) = \sqrt{\text{Det}(I-\bar{K})_{\mathbb{L}^2([s,+\infty[)}} := F^{(\epsilon)}(s),\tag{11}$$

---

$K_{12}(r',r) = \tilde{K}_{12}(-r,-r')\pi$ (using the change $(s,w)\to(w,-z)$), where $\tilde{K}$ denotes the kernel displayed in formula (5) and below in [25]. One has also performed a shift $r,r' \to r,r' + 2\log 2$, which together with the identification $\zeta/4 \equiv -\varsigma$ shows that $Z^{(0)}$ in [25] is identical to $e^H$ here. The extra $(-1)^{n_s} = (-1)^L$ factor amounts to the permutation of the column and lines of the $2\times 2$ block Pfaffian. Finally, $t/2$ there is $t$ here.

where the transition kernel for $\epsilon > 0$ is given by

$$\bar{K}(x,y) = \frac{1}{2} \iint_{C^2} \frac{\mathrm{d}w \mathrm{d}z}{(2i\pi)^2} \frac{\epsilon+w}{\epsilon-w} \frac{\epsilon+z}{\epsilon-z} \frac{w-z}{w+z} \frac{1}{w} e^{-xz-yw+\frac{w^3+z^3}{3}}. \tag{12}$$

Here, the contours $C$ are parallel to the imaginary axis and cross the real axis between 0 and $\epsilon$. For $\epsilon \to +\infty$, we check that this kernel converges towards the scalar kernel associated to the GSE, obtained in [24] for the droplet initial condition with $A = +\infty$. The limit $\epsilon \to 0^+$ is more delicate to handle, but via a careful analysis we are able to show that it converges towards the known scalar kernel of the GOE-TW distribution. We have introduced the notation $F^{(\epsilon)}(s)$ for further purpose. Hence we show here that $F^{(0)}(s) = F_1(s)$.

In Section 5.4 we extend our calculation and obtain the scalar kernel *at any finite time*. The explicit result in displayed in Eqs. (124), (125), (126) and (112).

Finally, although we do not address the case $A < -1/2$ or $\epsilon < 0$ the conjectured equivalence with the kernel of Ref. [37,38] (which holds for $\epsilon > 0$) suggests that for any $A < -\frac{1}{2}$, the one-point KPZ height has Gaussian fluctuations.

$$\lim_{t\to\infty} \mathbb{P}\left(\frac{h(0,t) + t(\frac{1}{12} - (A+\frac{1}{2})^2)}{t^{1/2}\sqrt{2A+1}} \leqslant s\right) = \frac{1}{\sqrt{2\pi}} \int_{-\infty}^{s} \mathrm{d}y\, e^{-\frac{y^2}{2}}, \tag{13}$$

which is also the conclusion of the RBA analysis of [29], arising there from the contribution of the boundary bound states. Physically it is expected indeed in the phase where the DP is bound to the wall.

## 3 Bethe ansatz calculation for Brownian initial condition with Dirichlet boundary condition

Let us recall that via the Cole-Hopf mapping the partition sum $Z(x,t) = e^{h(x,t)}$, where $h(x,t)$ is solution of the KPZ equation (1), satisfies the stochastic heat equation (SHE)

$$\partial_t Z(x,t) = \partial_x^2 Z(x,t) + \sqrt{2}\,\xi(x,t) Z(x,t) \tag{14}$$

understood here with the Ito prescription. Here we consider the problem on the half-line $x \geqslant 0$. Let us denote $Z_{\mathrm{Br}}(x,t)$ the solution of the SHE (14) with the Brownian initial condition in presence of a drift $A + \frac{1}{2}$

$$Z_{\mathrm{Br}}(x,0) = e^{B(x) - (A+\frac{1}{2})x}, \tag{15}$$

where $B(x)$ is the standard Brownian (i.e. with $B(0) = 0$), and with Dirichlet boundary condition $Z_{\mathrm{Br}}(x=0,t) = 0$. We will first calculate the integer moments of $Z_{\mathrm{Br}}(x,t)$, from which we will obtain the moments of the limit $\hat{Z}_{\mathrm{Br}}(t) = \lim_{\kappa\to 0^+} Z_{\mathrm{Br}}(\kappa,t)/\kappa$. Then we will use that

$$\mathbb{E}_{\mathrm{KPZ,B}}\left[\hat{Z}_{\mathrm{Br}}(t)^n\right] = \mathbb{E}_{\mathrm{KPZ}}\left[Z(t)^n\right] \tag{16}$$

to obtain the moments of $Z(t) = Z(x=0,t)$, which denotes here the solution of the SHE for the droplet initial condition and a wall of parameter $A$, our problem of main interest. Note that on the left hand side the average is over the Brownian $B(x)$ and the KPZ noise while on the r.h.s. it is over the KPZ noise only.

The general equal time moments of the solution of the SHE, $Z_{Br}(x, t)$, over the KPZ noise can be expressed [21] as quantum mechanical expectation values of the evolution operator in imaginary time of the Lieb Liniger model [59]

$$\mathbb{E}_{KPZ,B}\left[Z_{Br}(x_1, t)\ldots Z_{Br}(x_n, t)\right] = \langle x_1 \ldots x_n | e^{-tH_n} | \Psi(t=0)\rangle. \qquad (17)$$

Here $H_n$ is the Hamiltonian of the Lieb Liniger model [59] for $n$ quantum particles with attractive delta function interactions of strength $c = -\bar{c} < 0$

$$H_n = -\sum_{i=1}^n \partial_{x_i}^2 - 2\bar{c} \sum_{1 \leqslant i < j \leqslant n} \delta(x_i - x_j), \qquad (18)$$

with here an below, in our units $\bar{c} = 1$. The initial state is such that

$$\mathbb{E}_{KPZ,B}\left[Z_{Br}(x_1, t=0)\ldots Z_{Br}(x_n, t=0)\right] = \langle x_1 \ldots x_n | \Psi(t=0)\rangle. \qquad (19)$$

Since here we are considering the Brownian initial condition and interested in averages both over the Brownian and the KPZ noise we must take the initial state $|\Psi(t=0)\rangle$ as

$$\langle x_1 \ldots, x_n | \Psi(t=0)\rangle = \Phi_0(x_1, \ldots, x_n) := \mathbb{E}_B\left[\exp\left(\sum_{j=1}^n B(x_j) - (A + \frac{1}{2})x_j\right)\right]. \qquad (20)$$

A simple calculation shows that $\Phi_0(x_1, \ldots, x_n)$ is the fully symmetric function which in the sector $0 \leqslant x_1 < \cdots \leqslant x_n$ takes the form

$$\Phi_0(x_1, \ldots, x_n) = \exp\left(\sum_{j=1}^n \frac{1}{2}(2n - 2j + 1)x_j - (A + \frac{1}{2})x_j\right). \qquad (21)$$

### 3.1 Bethe ansatz formula for the moments

We can now rewrite (17) at coinciding points using the decomposition of the evolution operator $e^{-tH_n}$ in terms of the eigenstates of $H_n$ as

$$\mathbb{E}_{KPZ,B}\left[Z_{Br}(x, t)^n\right] = \sum_\mu \Psi_\mu(x, \ldots, x)\langle \Psi_\mu | \Phi_0\rangle \frac{1}{||\mu||^2} e^{-tE_\mu}. \qquad (22)$$

Here the un-normalized eigenfunctions of $H_n$ are denoted $\Psi_\mu$ (of norm denoted $||\mu||$) with eigenenergies $E_\mu$. Here we used the fact that only symmetric (i.e. bosonic) eigenstates contribute since the initial and final states are fully symmetric in the $x_i$. Hence the $\sum_\mu$ denotes a sum over all bosonic eigenstates of the Lieb-Liniger model, also called delta Bose gas, and $\langle \Psi_\mu | \Phi_0\rangle$ denotes the overlap, i.e. the hermitian scalar product of the initial state (21) with the eigenstate $\Psi_\mu$.

We should remember now that $H_n$ is defined on the half-line $x \geqslant 0$. The boundary condition (2) at the wall with parameter $A$ translates into the same boundary condition for the wavefunctions (in each of their coordinate). This half-line quantum mechanical problem can be solved by the Bethe ansatz for $A = +\infty$, i.e. for Dirichlet boundary condition as needed here [36, 60, 61] (see also section 5.1 of [62]). Note that it can also be solved for arbitrary $A$, [25, 62–68] which led to the formula in [28] and [29], but here we circumvent this, using instead the Brownian with Dirichlet boundary condition.

From the Bethe ansatz the eigenstates $\Psi_\mu$ are thus Bethe states, i.e. superpositions of plane waves over all permutations $P$ of the $n$ rapidities $\lambda_j$ for $j \in [1, n]$ with an additional summation over opposite pairs $\pm\lambda_j$ due to the infinite hard wall. The bosonic (fully symmetric) eigenstates can be obtained everywhere from their expression in the sector $x_1 < \cdots < x_n$, which reads

$$\Psi_\mu(x_1, \ldots, x_n) = \frac{1}{(2i)^n} \sum_{P \in S_n} \prod_{p=1}^n \left( \sum_{\varepsilon_p = \pm 1} \varepsilon_p e^{i\varepsilon_p x_p \lambda_{P(p)}} A[\varepsilon_1 \lambda_{P(1)}, \varepsilon_2 \lambda_{P(2)}, \ldots, \varepsilon_n \lambda_{P(n)}] \right),$$
$$A[\lambda_1, \ldots, \lambda_n] = \prod_{n \geq \ell > k \geq 1} (1 + \frac{i\bar{c}}{\lambda_\ell - \lambda_k})(1 + \frac{i\bar{c}}{\lambda_\ell + \lambda_k}). \tag{23}$$

This wavefunction automatically satisfies both

1. The matching condition arising from the $\delta(x_i - x_j)$ interaction

$$\left( \partial_{x_{i+1}} - \partial_{x_i} + \bar{c} \right) \Psi_\mu(x_1, \ldots, x_n) \big|_{x_{i+1} = x_i} = 0. \tag{24}$$

2. The hardwall boundary condition $\Psi_\mu(x_1, \ldots, x_n) = 0$ if some $x_i = 0$.

The allowed values for the rapidities $\lambda_i$, which parametrize the true physical eigenstates are determined by the Bethe equations arising from the boundary conditions at $x = L$ as discussed below. One will find that the normalized eigenstates $\psi_\mu = \Psi_\mu / ||\mu||$ vanish as $(\lambda_i - \lambda_j)$ or $(\lambda_i + \lambda_j)$ when two rapidities become equal or opposite: hence the rapidities obey an exclusion principle.

The detailed Bethe equations, which determine the allowed values for the set of rapidities $\{\lambda_j\}$, depend on the choice of boundary condition at $x = L$. However, in the $L \to +\infty$ limit, these details do not matter. For simplicity we choose another hardwall at $x = L$. The Bethe equations then read

$$e^{2i\lambda_j L} = \prod_{\ell \neq j} \frac{\lambda_j - \lambda_\ell - i\bar{c}}{\lambda_j - \lambda_\ell + i\bar{c}} \frac{\lambda_j + \lambda_\ell - i\bar{c}}{\lambda_j + \lambda_\ell + i\bar{c}}. \tag{25}$$

*Remark* 3.1. This set of equations is invariant by $\lambda_j \to -\lambda_j$ for any $j$.

In the case of the infinite hardwall, these equations are also given in Ref. [61] and their solutions in the large $L$ limit were studied in Ref. [69]. The structure of the states for infinite $L$ is found similar to the standard case, i.e. the general eigenstates are built by partitioning the $n$ particles into a set of $n_s$ bound-states formed by $m_j \geq 1$ particles with $n = \sum_{j=1}^{n_s} m_j$. Each bound state is a *perfect string* [70], i.e. a set of rapidities

$$\lambda^{j,a} = k_j + \frac{i\bar{c}}{2}(m_j + 1 - 2a), \tag{26}$$

where $a = 1, \ldots, m_j$ labels the rapidities within the string. Such eigenstates have momentum and energy

$$K_\mu = \sum_{j=1}^{n_s} m_j k_j, \qquad E_\mu = \sum_{j=1}^{n_s} m_j k_j^2 - \frac{\bar{c}^2}{12} m_j (m_j^2 - 1). \tag{27}$$

The ground-state corresponds to a single $n$-string with $k_1 = 0$. The difference with the standard case is that the states are now invariant by a sign change of any of the momenta $\lambda_j \to -\lambda_j$,

i.e. $k_j \to -k_j$. From now on, we will denote the wavefunctions of the string states as $\Psi_{\{k_\ell, m_\ell\}}$. The inverse of the squared norm of an arbitrary string state was obtained in Ref. [24] as [2]

$$
\begin{aligned}
\||\mu\||^2 &:= \int_0^L dx_1 \cdots \int_0^L dx_n |\Psi_{\{k_\ell, m_\ell\}}(x_1, \ldots, x_n)|^2 \\
\frac{1}{\||\mu\||^2} &= \frac{1}{n!} \bar{c}^{n-n_s} 2^{n_s} \prod_{i=1}^{n_s} S_{k_i, m_i} \prod_{1 \leqslant i < j \leqslant n_s} D_{k_i, m_i, k_j, m_j} L^{-n_s} \\
D_{k_1, m_1, k_2, m_2} &= \left( \frac{4(k_1 - k_2)^2 + (m_1 - m_2)^2 c^2}{4(k_1 - k_2)^2 + (m_1 + m_2)^2 c^2} \right) \times \left( \frac{4(k_1 + k_2)^2 + (m_1 - m_2)^2 c^2}{4(k_1 + k_2)^2 + (m_1 + m_2)^2 c^2} \right) \\
S_{k,m} &= \frac{2^{2m-2}}{m^2} \prod_{p=1}^{[m/2]} \frac{4k^2 + c^2(m - 2p)^2}{4k^2 + c^2(m + 1 - 2p)^2},
\end{aligned}
\tag{28}
$$

with $S_{k,1} = 1$. Note that we have only kept the leading term in $L$ as $L \to +\infty$. Inserting the norm formula Eq. (28) into Eq. (22), we obtain the starting formula for the integer moments of the partition sum with Brownian weight on the endpoint in the limit $L \to +\infty$

$$
\begin{aligned}
\mathbb{E}_{\mathrm{KPZ,B}}\left[ Z_{\mathrm{Br}}(x,t)^n \right] = &\sum_{n_s=1}^n \frac{2^{n_s} \bar{c}^n}{n_s! \bar{c}^n n!} \prod_{p=1}^{n_s} \sum_{m_p \geqslant 1} \int_{\mathbb{R}} \frac{dk_p}{2\pi} m_p S_{k_p, m_p} e^{(m_p^3 - m_p) \frac{\bar{c}^2 t}{12} - m_p k_p^2 t} \\
&\times \delta_{n, \sum_{j=1}^{n_s} m_j} \prod_{i<j}^{n_s} D_{k_i, m_i, k_j, m_j} \Psi_{\{k_\ell, m_\ell\}}(x, \ldots, x) \langle \Psi_{\{k_\ell, m_\ell\}} | \Phi_0 \rangle.
\end{aligned}
\tag{29}
$$

Here the Kronecker delta enforces the constraint $\sum_{j=1}^{n_s} m_j = n$ with $m_j \geqslant 1$ and in the summation over states we used $\sum_{k_j} \to m_j L \int_{\mathbb{R}} \frac{dk}{2\pi}$ which holds also here in the large $L$ limit: the momenta sums become continuous and one can use that the string momenta $m_j k_j$ correspond to free particles as in Refs. [9, 12, 13, 24, 29]. Since we are interested in

$$
\hat{Z}_{\mathrm{Br}}(t) = \lim_{x \to 0^+} \frac{Z_{\mathrm{Br}}(x,t)}{x},
\tag{30}
$$

we can simplify the factor $\Psi_{\{k_\ell, m_\ell\}}(x, \cdots, x)$ in (29). For the general Bethe state (23) (before insertion of the string solution), the small $x$ limit reads

$$
\Psi_\mu(x, \ldots, x) = n! x^n \prod_{j=1}^n \lambda_j + \mathcal{O}(x^{n+1}).
\tag{31}
$$

Inserting the string solution we see that we can replace in (29) at small $x$

$$
\Psi_{\{k_\ell, m_\ell\}}(x, \cdots, x) = n! x^n \prod_{j=1}^{n_s} A_{k_j, m_j} + \mathcal{O}(x^{n+1}),
\tag{32}
$$

$$
A_{k,m} = \prod_{a=1}^m (k + i\frac{\bar{c}}{2}(m + 1 - 2a)) = (-i\bar{c})^m \frac{\Gamma(\frac{1+m}{2} + \frac{ik}{\bar{c}})}{\Gamma(\frac{1-m}{2} + \frac{ik}{\bar{c}})}.
\tag{33}
$$

To obtain the $n$-th moment of (30) from (29) we still need to calculate the overlap $\langle \Psi_{\{k_\ell, m_\ell\}} | \Phi_0 \rangle$ where $\Phi_0$ is given in (21). In general it involves sums over permutations and leads to complicated expressions unless there is some kind of "miracle", known to happen in full space only for a few special initial conditions (droplet, half-flat, Brownian). Here, as we

---

[2]correcting the small misprint in formula (9) in [24].

find in the Appendix A, the final result in the half-space for Brownian initial conditions is quite simple

$$\langle \Psi_\mu | \Phi_0 \rangle = n! \prod_{j=1}^{n} \frac{\lambda_j}{A^2 + \lambda_j^2}. \tag{34}$$

Inserting the rapidities $\lambda_j$ of the string state one see that the denominator of this formula reads

$$E_{k,j} = \prod_{a=1}^{m} \frac{1}{A^2 + (k + i\frac{\bar{c}}{2}(m+1-2a))^2} = \frac{1}{\bar{c}^{2m}} \frac{\Gamma(\frac{1-m}{2} + \frac{A+ik}{\bar{c}})\,\Gamma(\frac{1-m}{2} + \frac{A-ik}{\bar{c}})}{\Gamma(\frac{1+m}{2} + \frac{A+ik}{\bar{c}})\,\Gamma(\frac{1+m}{2} + \frac{A-ik}{\bar{c}})}, \tag{35}$$

while the numerator was already calculated in (32). We can thus define $C_{k,j} = A_{k,j}^2 E_{k,j}$ and putting all together we obtain the starting expression for the integer moments:

$$\mathbb{E}_{\text{KPZ}}\left[\hat{Z}_{\text{Br}}(t)^n\right] = \sum_{n_s=1}^{n} \frac{2^{n_s} \bar{c}^n n!}{n_s! \bar{c}^{n_s}} \quad \prod_{p=1}^{n_s} \sum_{m_p \geqslant 1} \int_{\mathbb{R}} \frac{dk_p}{2\pi} C_{k_p, m_p} m_p S_{k_p, m_p} e^{(m_p^3 - m_p)\frac{\bar{c}^2 t}{12} - m_p k_p^2 t}$$

$$\times \delta_{n, \sum_{j=1}^{n_s} m_j} \prod_{i<j}^{n_s} D_{k_i, m_i, k_j, m_j}, \tag{36}$$

where the expressions for $C, S, D$ are given above and where we recall the constraint $\sum_{j=1}^{n_s} m_j = n$. Let us use $\bar{c} = 1$ from now on. Denoting

$$B_{k,m} = 4m^2 C_{k,m} S_{k,m}$$

$$= 2k(-i)^{2m-1} \prod_{j=1-m}^{m-1} (2ik+j) \prod_{a=1}^{m} \frac{1}{A^2 + (k + i\frac{1}{2}(m+1-2a))^2}$$

$$= \prod_{j=0}^{m-1} (4k^2 + j^2) \frac{\Gamma(\frac{1-m}{2} + A + ik)}{\Gamma(\frac{1+m}{2} + A + ik)} \frac{\Gamma(\frac{1-m}{2} + A - ik)}{\Gamma(\frac{1+m}{2} + A - ik)} \tag{37}$$

$$= \frac{2k}{\pi} \sinh(2\pi k) \Gamma(2ik+m) \Gamma(-2ik+m) \frac{\Gamma(\frac{1-m}{2} + A + ik)}{\Gamma(\frac{1+m}{2} + A + ik)} \frac{\Gamma(\frac{1-m}{2} + A - ik)}{\Gamma(\frac{1+m}{2} + A - ik)}.$$

The starting formula for the moments of our two equivalent problems (droplet initial conditions with any $A$, and Brownian initial condition with drift $A + 1/2$ and Dirichlet) reads

$$\mathbb{E}_{\text{KPZ}}\left[Z(t)^n\right] = \mathbb{E}_{\text{KPZ,B}}\left[\hat{Z}_{\text{Br}}(t)^n\right]$$

$$= \sum_{n_s=1}^{n} \frac{n! 2^{n_s}}{n_s!} \prod_{p=1}^{n_s} \sum_{m_p \geqslant 1} \int_{\mathbb{R}} \frac{dk_p}{2\pi} \frac{B_{k_p, m_p}}{4m_p} e^{(m_p^3 - m_p)\frac{t}{12} - m_p k_p^2 t} \delta_{n, \sum_{j=1}^{n_s} m_j} \prod_{i<j}^{n_s} D_{k_i, m_i, k_j, m_j}, \tag{38}$$

where $B_{k,m}$ is given in (37) and $D_{k_i, m_i, k_j, m_j}$ is given in (28) and where we recall the constraint $\sum_{j=1}^{n_s} m_j = n$. This formula is identical to the one obtained by a completely different calculation in [28] for the problem of the droplet initial condition with any $A$, consistent with the theorem of Ref. [40]. For $A \to +\infty$ one recovers the expression (11) in [24], i.e the same without the ratio of Gamma functions involving $A$, using that $Z$ there is $\lim_{A \to +\infty} A^2 Z(t)$ here.

It is important to note that, strictly, the formula (34) for the overlap of the string states with the Brownian initial condition is valid only when the multiple integrals in the scalar product are convergent. This requires (see discussion in the Appendix A) the condition $n/2 < A + \frac{1}{2}$. Hence the above formula for the integer moments $\mathbb{E}_{\text{KPZ,B}}\left[\hat{Z}_{\text{Br}}(t)^n\right]$ is valid only when the drift

is large enough, for each value of $n$. This requirement for the drift is identical to the one obtained in the full space, for the Brownian initial condition in [14,15] and for the half-Brownian initial condition in [71]. There, it was shown how to nevertheless use these restricted moment formula to construct the full solution for the generating function of the moments at finite time (see definition below). Here we follow the same strategy as in these works and we refer the reader to Ref. [15] Section 4.3 for further details. The main idea behind the continuation of the moment formula is that it is possible to avoid all divergences in the complex plane by finely tuning the contour integrals of the Mellin-Barnes summation formula, which is possible as long as $A + 1/2 \geqslant 0$.

The resulting solution obtained in [14, 15, 71] for the full space holds for any positive drifts. The limit of zero drift is quite delicate, but can be performed, and leads to the Baik Rains distribution for stationary KPZ. The analog in the half-space is that we obtain below a formula valid for positive drift, $A > -1/2$. With some efforts, we are able to extend it to $A = -1/2$ the critical case, leading to the GOE-TW distribution. We have not attempted to obtain the solution for $A < -1/2$, i.e. negative drift.

Since the two problems are in correspondence, due to the theorem of Parekh in [40], the above formula for the moments $\mathbb{E}_{\mathrm{KPZ}}\big[Z(t)^n\big]$ for the droplet initial condition and wall parameter $A$ is also valid only for $n/2 < A + \frac{1}{2}$. In that case, this restriction arises from an a priori different origin. As shown in [29], the string states discussed above do not form a complete basis for generic $A$: new boundary bound states arise. To obtain a formula for the moments valid for any $n,A$ is possible but requires to include all boundary states, as done in [29]. This is illustrated here in the Appendix B for the first moment of $Z(t)$. We will not follow that route here, although connections between the two approaches must exist.

## 3.2 Moments in terms of a Pfaffian

An important identity, which makes the problem solvable in the end, is that the inverse norms of the states can be expressed as a Schur Pfaffian. Introducing the reduced variables $X_{2p-1} = m_p + 2ik_p$ and $X_{2p} = m_p - 2ik_p$ for $p \in [1, n_s]$, the norm reads

$$\prod_{1 \leqslant i < j \leqslant n_s} D_{k_i, m_i, k_j, m_j} = \prod_{j=1}^{n_s} \frac{m_j}{2ik_j} \Pf_{2n_s \times 2n_s} \left[ \frac{X_i - X_j}{X_i + X_j} \right], \tag{39}$$

where we recall that the Pfaffian of an anti-symmetric matrix $A$ of size $N \times N$ is defined by

$$\Pf(A) = \sqrt{\mathrm{Det}(A)} = \sum_{\substack{\sigma \in S_N, \\ \sigma(2p-1) < \sigma(2p)}} \mathrm{sign}(\sigma) \prod_{p=1}^{N/2} A_{\sigma(2p-1), \sigma(2p)}, \tag{40}$$

and that the Schur Pfaffian is given by (see Ref. [78])

$$\Pf\left[ \frac{X_i - X_j}{X_i + X_j} \right] = \prod_{i<j} \frac{X_i - X_j}{X_i + X_j}. \tag{41}$$

Hence the starting formula for the moments now becomes:

$$\mathbb{E}_{\mathrm{KPZ}}\big[Z(t)^n\big] = \mathbb{E}_{\mathrm{KPZ,B}}\big[\hat{Z}_{\mathrm{Br}}(t)^n\big]$$
$$= \sum_{n_s=1}^{n} \frac{n!}{n_s!} \prod_{p=1}^{n_s} \sum_{m_p \geqslant 1} \int_{\mathbb{R}} \frac{dk_p}{2\pi} \frac{B_{k_p, m_p}}{4ik_p} e^{(m_p^3 - m_p)\frac{t}{12} - m_p k_p^2 t} \delta_{n, \sum_{j=1}^{n_s} m_j} \Pf_{2n_s \times 2n_s} \left[ \frac{X_i - X_j}{X_i + X_j} \right]. \tag{42}$$

### 3.3  Generating function in terms of a Fredholm Pfaffian

We will now write the generating function for the moments of $Z(t)$, i.e. focusing on the droplet initial condition with generic $A$ (the result being identical for the Brownian with Dirichlet boundary conditions). It is defined, for $\varsigma > 0$, as

$$g(\varsigma) = \mathbb{E}_{\mathrm{KPZ}}\left[\exp(-\varsigma e^{\frac{t}{12}} Z(t))\right] = 1 + \sum_{n=1}^{\infty} \frac{(-\varsigma e^{\frac{t}{12}})^n}{n!} \mathbb{E}_{\mathrm{KPZ}}\left[Z(t)^n\right]. \tag{43}$$

The constraint $\sum_{i=1}^{n_s} m_i = n$ in Eq. (42) can then be relaxed by reorganizing the series according to the number of strings:

$$g(\varsigma) = 1 + \sum_{n_s=1}^{\infty} \frac{1}{n_s!} Z(n_s, \varsigma), \tag{44}$$

where $Z(n_s, \varsigma)$ is the partition sum at fixed number of strings, calculated below. We now show that one can write the generating function as a Fredholm Pfaffian. It will be possible thanks to the Schur Pfaffian identity, Eq. (39), given above. The partition sum at fixed number of strings, expressed in terms of the reduced variables $X_{2p-1} = m_p + 2ik_p$ and $X_{2p} = m_p - 2ik_p$ for $p \in [1, n_s]$, reads

$$Z(n_s, \varsigma) = \prod_{p=1}^{n_s} \sum_{m_p \geqslant 1} \int_{\mathbb{R}} \frac{dk_p}{2\pi} (-\varsigma)^{m_p} \frac{B_{k_p, m_p}}{4ik_p} e^{-tm_p k_p^2 + \frac{t}{12} m_p^3} \underset{2n_s \times 2n_s}{\mathrm{Pf}}\left[\frac{X_i - X_j}{X_i + X_j}\right], \tag{45}$$

where $B_{k,m}$ was given in (37). The summation over the variables $m_p$ can be done using the Mellin-Barnes summation trick similarly to Refs. [14,15]. The barrier $A > (n-1)/2$ is overcome exactly as in Ref. [15] (see Lemma. 6 and the discussion following there) from an analytic continuation of Gamma functions included in the $B_{k,m}$ factor, the introduction of a particular contour $C_0$ and a final requirement for the drift $A + 1/2 > 0$. Indeed, define the contour $C_0 = a + i\mathbb{R}$ with $a \in ]0, \min(2A+1, 1)[$, then denoting the summand of Eq. (45) by the function $f(m_p)$, we have

$$\sum_{m \geqslant 1} (-\varsigma)^m f(m) = -\int_{C_0} \frac{dw}{2i\pi} \varsigma^w \frac{\pi}{\sin \pi w} f(w) = -\int_{\mathbb{R}} dr \frac{\varsigma}{\varsigma + e^{-r}} \int_{C_0} \frac{dw}{2i\pi} f(w) e^{-wr}. \tag{46}$$

For each $m_p$ we therefore introduce two variables $r_p$ and $w_p$ and we redefine the reduced variables $X_{2p}$ and $X_{2p-1}$ under the minimal replacement $m_p \to w_p$ imposed by the Mellin-Barnes formula. This leads to the following rewriting of the partition sum at fixed number of strings (see section 5 in [27] for similar manipulations)

$$\begin{aligned} Z(n_s, \varsigma) = (-1)^{n_s} \prod_{p=1}^{n_s} & \int_{\mathbb{R}} dr_p \frac{\varsigma}{\varsigma + e^{-r_p}} \iint_{C_0^2} \frac{dX_{2p-1}}{4i\pi} \frac{dX_{2p}}{4i\pi} \frac{\sin(\frac{\pi}{2}(X_{2p} - X_{2p-1}))}{2\pi} \\ & \times \Gamma(X_{2p-1})\Gamma(X_{2p}) \frac{\Gamma(A + \frac{1}{2} - \frac{X_{2p}}{2})}{\Gamma(A + \frac{1}{2} + \frac{X_{2p}}{2})} \frac{\Gamma(A + \frac{1}{2} - \frac{X_{2p-1}}{2})}{\Gamma(A + \frac{1}{2} + \frac{X_{2p-1}}{2})} \\ & \times e^{-(X_{2p-1} + X_{2p})\frac{r_p}{2} + t(\frac{X_{2p-1}^3}{24} + \frac{X_{2p}^3}{24})} \underset{2n_s \times 2n_s}{\mathrm{Pf}}\left[\frac{X_i - X_j}{X_i + X_j}\right]. \end{aligned} \tag{47}$$

*Remark* 3.2. Note that the contour $C_0$ passes to the left of the pole of the Gamma function at $X = 2A + 1$.

We observe that the integrals are almost separable in $X_{2p}$ and $X_{2p-1}$ except for the sine function which couples them. By anti-symmetrization and similarly to Ref. [27] section 5, we can proceed to the replacement

$$\sin(\frac{\pi}{2}(X_{2p} - X_{2p-1})) \to 2\sin(\frac{\pi}{2}X_{2p})\cos(\frac{\pi}{2}X_{2p-1}). \tag{48}$$

The last manipulations consist in rescaling all variables $X$ by a factor 2 and replacing the contours of integration by $C = \frac{a}{2} + i\mathbb{R}$. Hence we have

$$
\begin{aligned}
Z(n_s, \varsigma) =& (-1)^{n_s} \prod_{p=1}^{n_s} \int_{\mathbb{R}} dr_p \frac{\varsigma}{\varsigma + e^{-r_p}} \iint_{C^2} \frac{dX_{2p-1}}{2i\pi} \frac{dX_{2p}}{2i\pi} \frac{\sin(\pi X_{2p})\cos(\pi X_{2p-1})}{\pi} \\
& \times \Gamma(2X_{2p-1})\Gamma(2X_{2p}) \frac{\Gamma(A + \frac{1}{2} - X_{2p})}{\Gamma(A + \frac{1}{2} + X_{2p})} \frac{\Gamma(A + \frac{1}{2} - X_{2p-1})}{\Gamma(A + \frac{1}{2} + X_{2p-1})} \\
& \times e^{-(X_{2p-1} + X_{2p})r_p + t(\frac{X_{2p-1}^3}{3} + \frac{X_{2p}^3}{3})} \Pf_{2n_s \times 2n_s} \left[ \frac{X_i - X_j}{X_i + X_j} \right].
\end{aligned}
\tag{49}
$$

There are a few last steps before we introduce the Fredholm Pfaffian. First define the functions

$$
\begin{aligned}
\phi_{2p}(X) &= \frac{\sin(\pi X)}{\pi} \Gamma(2X) \frac{\Gamma(A + \frac{1}{2} - X)}{\Gamma(A + \frac{1}{2} + X)} e^{-r_p X + t\frac{X^3}{3}}, \\
\phi_{2p-1}(X) &= \cos(\pi X)\Gamma(2X) \frac{\Gamma(A + \frac{1}{2} - X)}{\Gamma(A + \frac{1}{2} + X)} e^{-r_p X + t\frac{X^3}{3}}.
\end{aligned}
\tag{50}
$$

Using a known property of Pfaffians (see De Bruijn [72]), we can rewrite the partition sum at fixed number of strings itself as a Pfaffian, i.e. we use that

$$
\prod_{\ell=1}^{2n_s} \int_C \frac{dX_\ell}{2i\pi} \Phi_\ell(X_\ell) \Pf_{2n_s \times 2n_s} \left[ \frac{X_i - X_j}{X_i + X_j} \right] = \Pf_{2n_s \times 2n_s} \left[ \iint_{C^2} \frac{dw}{2i\pi} \frac{dz}{2i\pi} \Phi_i(w)\Phi_j(z) \frac{w - z}{w + z} \right]. \tag{51}
$$

This leads to the definition of a $2n_s \times 2n_s$ matrix $M$ such that

$$
M_{ij} = \iint_{C^2} \frac{dw}{2i\pi} \frac{dz}{2i\pi} \Phi_i(w)\Phi_j(z) \frac{w - z}{w + z}. \tag{52}
$$

Since a variable $r_p$ will be shared between four elements of this matrix, it is more convenient to view $M$ as composed of $2 \times 2$ blocks which we denote $K$, whose elements are presented in Eqs. (7) (for visualization see formula (146)). Finally, the string-replicated partition function is given by an infinite series of Pfaffians

$$
g(\varsigma) = 1 + \sum_{n_s=1}^{\infty} \frac{(-1)^{n_s}}{n_s!} \prod_{p=1}^{n_s} \int_{\mathbb{R}} dr_p \frac{\varsigma}{\varsigma + e^{-r_p}} \Pf_{n_s \times n_s} [K(r_k, r_\ell)]. \tag{53}
$$

This series admits a closed form in terms of a Fredholm Pfaffian, which is our main result for the generating function at finite time together with the explicit expression of the kernel given in Eqs. (7)

$$
g(\varsigma) = \mathbb{E}_{KPZ} \left[ \exp(-\varsigma e^{H(t)}) \right] = \Pf(J - \sigma_\varsigma K)_{\mathbb{L}^2(\mathbb{R})}. \tag{54}
$$

The function $\sigma_\varsigma$ is given by $\sigma_\varsigma(r) = \frac{\varsigma}{\varsigma + e^{-r}}$ and the $2 \times 2$ kernel $J$ is given by $J(r, r') = \begin{pmatrix} 0 & 1 \\ -1 & 0 \end{pmatrix} \mathbb{1}_{r=r'}$. For the precise definition and properties of Fredholm Pfaffians see Section 8 in [73], as well as e.g. Section 2.2. in [37], Appendix B in [74] and Appendix G in [12, 13].

## 3.4 An equivalent kernel

For the antisymmetrization, we used the trigonometric decomposition

$$\sin(\pi(X_{2p}-X_{2p-1})) = \sin(\pi X_{2p})\cos(\pi X_{2p-1}) - \cos(\pi X_{2p})\sin(\pi X_{2p-1}). \tag{55}$$

The decomposition can be made more general, and for later purpose, for any real $\alpha$, we decompose the sine function as

$$\sin(\pi(X_{2p}-X_{2p-1})) = \sin(\pi(X_{2p}-\alpha))\cos(\pi(X_{2p-1}-\alpha)) - \cos(\pi(X_{2p}-\alpha))\sin(\pi(X_{2p-1}-\alpha)). \tag{56}$$

For the rest of the paper we will call this the *$\alpha$-decomposition*. To study the limit $A \to +\infty$, we will use $\alpha = 0$ and for the limit $A \to -\frac{1}{2}$, we will use $\alpha = A + \frac{1}{2}$. In particular, from the same symmetrization argument than above, the partition function for $n_s$ strings reads, using the $\alpha$-decomposition:

$$\begin{aligned}
Z(n_s,\varsigma) = & (-1)^{n_s} \prod_{p=1}^{n_s} \int_{\mathbb{R}} dr_p \frac{\varsigma}{\varsigma + e^{-r_p}} \iint_{C^2} \frac{dX_{2p-1}}{2i\pi} \frac{dX_{2p}}{2i\pi} \frac{\sin(\pi(X_{2p}-\alpha))\cos(\pi(X_{2p-1}-\alpha))}{\pi} \\
& \times \Gamma(2X_{2p-1})\Gamma(2X_{2p}) \frac{\Gamma(A+\frac{1}{2}-X_{2p})}{\Gamma(A+\frac{1}{2}+X_{2p})} \frac{\Gamma(A+\frac{1}{2}-X_{2p-1})}{\Gamma(A+\frac{1}{2}+X_{2p-1})} \\
& \times e^{-(X_{2p-1}+X_{2p})r_p + t(\frac{X_{2p-1}^3}{3}+\frac{X_{2p}^3}{3})} \underset{2n_s \times 2n_s}{\text{Pf}} \left[ \frac{X_i-X_j}{X_i+X_j} \right].
\end{aligned} \tag{57}$$

This leads to an $\alpha$-extension of our functions $\phi$, which in the special case $\alpha = A + \frac{1}{2}$ read

$$\begin{aligned}
\phi_{2p}(X) &= \frac{\sin(\pi(X-(A+\frac{1}{2})))}{\pi} \Gamma(2X) \frac{\Gamma(A+\frac{1}{2}-X)}{\Gamma(A+\frac{1}{2}+X)} e^{-r_p X + t\frac{X^3}{3}}, \\
\phi_{2p-1}(X) &= \cos(\pi(X-(A+\frac{1}{2}))) \Gamma(2X) \frac{\Gamma(A+\frac{1}{2}-X)}{\Gamma(A+\frac{1}{2}+X)} e^{-r_p X + t\frac{X^3}{3}}.
\end{aligned} \tag{58}$$

We see that this choice $\alpha = A + \frac{1}{2}$ can be interesting since it suppresses the pole at $X = A + \frac{1}{2}$ in the function $\phi_{2p}$, a property that we will use below. At the end we obtain a the same Fredholm Pfaffian expression for the generating function, with the minimal replacement in the kernel (7)

$$\begin{aligned}
\sin(\pi X) &\to \sin(\pi(X-(A+\frac{1}{2}))), \\
\cos(\pi X) &\to \cos(\pi(X-(A+\frac{1}{2}))).
\end{aligned} \tag{59}$$

We emphasize that all the kernels parametrized by $\alpha$ yield the same Fredholm Pfaffian by construction.

# 4  Large time limit of the Fredholm Pfaffian and the distribution of the KPZ height

We will now study the large time limit of our kernel. To understand the scaling required at large time, let us recall the expression of the partition sum at fixed number of strings

$$
\begin{aligned}
Z(n_s, \varsigma) =& (-1)^{n_s} \prod_{p=1}^{n_s} \int_{\mathbb{R}} dr_p \frac{\varsigma}{\varsigma + e^{-r_p}} \iint_{C^2} \frac{dX_{2p-1}}{2i\pi} \frac{dX_{2p}}{2i\pi} \frac{\sin(\pi X_{2p}) \cos(\pi X_{2p-1})}{\pi} \\
& \times \Gamma(2X_{2p-1}) \Gamma(2X_{2p}) \frac{\Gamma(A + \frac{1}{2} - X_{2p})}{\Gamma(A + \frac{1}{2} + X_{2p})} \frac{\Gamma(A + \frac{1}{2} - X_{2p-1})}{\Gamma(A + \frac{1}{2} + X_{2p-1})} \\
& \times e^{-(X_{2p-1} + X_{2p})r_p + t\left(\frac{X_{2p-1}^3}{3} + \frac{X_{2p}^3}{3}\right)} \underset{2n_s \times 2n_s}{\mathrm{Pf}} \left[ \frac{X_i - X_j}{X_i + X_j} \right].
\end{aligned}
\tag{60}
$$

At large time, we want to eliminate the time factor in the exponential, hence we perform the change of variables

$$
X = t^{-1/3} \tilde{X}, \qquad r = t^{1/3} \tilde{r}, \qquad A + \frac{1}{2} = \epsilon t^{-1/3}.
\tag{61}
$$

In the large time limit, the Gamma, cosine and sine functions simplify using that for small positive argument

$$
\Gamma(x) \simeq \frac{1}{x}, \qquad \cos(x) \simeq 1, \qquad \sin(x) \simeq x.
\tag{62}
$$

Under these simplifications, the partition sum at fixed number of strings reads in the limit $t \to +\infty$ (dropping all tildes)

$$
\begin{aligned}
Z(n_s, \varsigma) =& (-1)^{n_s} \prod_{p=1}^{n_s} \int_{\mathbb{R}} dr_p \frac{\varsigma}{\varsigma + e^{-t^{1/3} r_p}} \iint_{C^2} \frac{dX_{2p-1}}{2i\pi} \frac{dX_{2p}}{2i\pi} \frac{1}{4X_{2p-1}} \frac{\epsilon + X_{2p}}{\epsilon - X_{2p}} \frac{\epsilon + X_{2p-1}}{\epsilon - X_{2p-1}} \\
& \times e^{-(X_{2p-1} + X_{2p})r_p + \frac{X_{2p-1}^3}{3} + \frac{X_{2p}^3}{3}} \underset{2n_s \times 2n_s}{\mathrm{Pf}} \left[ \frac{X_i - X_j}{X_i + X_j} \right].
\end{aligned}
\tag{63}
$$

The contours $C$ have now to be understood as $C = \tilde{a} + i\mathbb{R}$, where $\tilde{a} \in ]0, \epsilon[$. We emphasize that the contours all lie at the left of the poles at $X = \epsilon$. Now that the rescaling at large time is well understood, we can reconstruct the matrix valued kernel $K^\epsilon$ which reads in this limit

$$
\begin{aligned}
K_{11}^\epsilon(r, r') &= \frac{1}{4} \iint_{C^2} \frac{dw}{2i\pi} \frac{dz}{2i\pi} \frac{w - z}{w + z} \frac{1}{wz} \frac{\epsilon + w}{\epsilon - w} \frac{\epsilon + z}{\epsilon - z} e^{-rw - r'z + \frac{w^3 + z^3}{3}}, \\
K_{22}^\epsilon(r, r') &= \frac{1}{4} \iint_{C^2} \frac{dw}{2i\pi} \frac{dz}{2i\pi} \frac{w - z}{w + z} \frac{\epsilon + w}{\epsilon - w} \frac{\epsilon + z}{\epsilon - z} e^{-rw - r'z + \frac{w^3 + z^3}{3}}, \\
K_{12}^\epsilon(r, r') &= \frac{1}{4} \iint_{C^2} \frac{dw}{2i\pi} \frac{dz}{2i\pi} \frac{w - z}{w + z} \frac{1}{w} \frac{\epsilon + w}{\epsilon - w} \frac{\epsilon + z}{\epsilon - z} e^{-rw - r'z + \frac{w^3 + z^3}{3}}.
\end{aligned}
\tag{64}
$$

*Remark* 4.1. The kernel $K^\epsilon$ has a particular structure, indeed its elements are related through derivative identities: $K_{22}^\epsilon(r, r') = \partial_r \partial_{r'} K_{11}^\epsilon(r, r')$, $K_{12}^\epsilon(r, r') = -\partial_{r'} K_{11}^\epsilon(r, r')$ and $K_{22}^\epsilon(r, r') = -\partial_r K_{12}^\epsilon(r, r')$.

*Remark* 4.2. The kernel $K^\epsilon$ can be obtained equivalently from the kernel (7) by the same rescaling of $r_p = t^{1/3} \tilde{r}_p$ and the changes $(w, z) \to t^{-1/3}(w, z)$ in the integrals.

One has $K_{11}(t^{1/3}\tilde{r}, t^{1/3}\tilde{r}') = K_{11}^\epsilon(\tilde{r}, \tilde{r}')$, $K_{22}(t^{1/3}\tilde{r}, t^{1/3}\tilde{r}') = t^{-2/3}K_{22}^\epsilon(\tilde{r}, \tilde{r}')$, $K_{12}(t^{1/3}\tilde{r}, t^{1/3}\tilde{r}') = t^{-1/3}K_{12}^\epsilon(\tilde{r}, \tilde{r}')$. This produces a factor $t^{-n_s/3}$ from the Pfaffian, compensating for the change of measure $\prod_p dr_p = t^{n_s/3} \prod_p d\tilde{r}_p$.

Finally, choosing the variable $\varsigma$ as $\varsigma = e^{-st^{1/3}}$, at large time we have

$$\lim_{t\to+\infty} \sigma_\varsigma(rt^{1/3}) = \Theta(r-s), \tag{65}$$

where $\Theta$ is the Theta Heaviside function. The Fredholm Pfaffian formula for the generating function then becomes in the limit

$$\lim_{t\to+\infty} g(\varsigma = e^{-st^{1/3}}) = \mathrm{Pf}(J - K^\epsilon)_{\mathbb{L}^2([s,+\infty[)}. \tag{66}$$

On the other hand, at large time, the Laplace transform of the distribution of the exponential of the KPZ height converges towards the cumulative probability of the height, i.e.

$$\begin{aligned}
\mathbb{E}_{\mathrm{KPZ}}\Big[\exp(-\varsigma e^{H(t)})\Big] &= \mathbb{E}_{\mathrm{KPZ}}\Big[\exp(-e^{H(t)-st^{1/3}})\Big] \\
&\simeq_{t\to+\infty} \mathbb{E}_{\mathrm{KPZ}}\Big[\Theta(st^{1/3} - H(t))\Big] \\
&\simeq_{t\to+\infty} \mathbb{P}(\frac{H(t)}{t^{1/3}} \leqslant s),
\end{aligned} \tag{67}$$

where $\Theta$ is the Theta Heaviside function. From this, we obtain the CDF of the height distribution in the large time limit as a Fredholm Pfaffian

$$\lim_{t\to+\infty} \mathbb{P}(\frac{H(t)}{t^{1/3}} \leqslant s) = \mathrm{Pf}(J - K^\epsilon)_{\mathbb{L}^2([s,+\infty[)} := F^{(\epsilon)}(s) \tag{68}$$

in terms of the matrix kernel given in (64), also called the transition kernel, as it describes the critical region around the transition. Anticipating a bit, it should describe the crossover from GSE/GOE/Gaussian discussed in the introduction, expected from universality arguments. Although we will indeed show most of it, our formula for the kernel is, however, limited at this stage to $\epsilon > 0$.

## 4.1 Large $\epsilon$ behavior of the matrix valued kernel and convergence to the GSE-TW distribution

Performing the large time limit at any fixed $A > -1/2$ corresponds to the previous calculation with $\epsilon = +\infty$. Hence we now investigate the large $\epsilon$ behavior of the kernel at large time. As the contours of kernel $K^\epsilon$ are parallel to the imaginary axis and cross the real axis between 0 and $\epsilon$, we can push the limit $\epsilon \to \infty$ without any ambiguity. All rational functions involving the parameter $\epsilon$ in the large time limit of the kernel $K^\epsilon$ in Eq. (64) converge to the value $-1$. Hence in this limit we obtain a kernel which we denote $K^\infty$, and it reads

$$\begin{aligned}
K_{11}^\infty(r, r') &= \frac{1}{4} \iint_{C^2} \frac{dw}{2i\pi} \frac{dz}{2i\pi} \frac{w-z}{w+z} \frac{1}{wz} e^{-rw-r'z+\frac{w^3+z^3}{3}}, \\
K_{22}^\infty(r, r') &= \frac{1}{4} \iint_{C^2} \frac{dw}{2i\pi} \frac{dz}{2i\pi} \frac{w-z}{w+z} e^{-rw-r'z+\frac{w^3+z^3}{3}}, \\
K_{12}^\infty(r, r') &= \frac{1}{4} \iint_{C^2} \frac{dw}{2i\pi} \frac{dz}{2i\pi} \frac{w-z}{w+z} \frac{1}{w} e^{-rw-r'z+\frac{w^3+z^3}{3}}.
\end{aligned} \tag{69}$$

This is precisely the kernel associated to the Gaussian Symplectic Ensemble (GSE) of random matrices as given in Lemma 2.7. of Ref. [37]. Hence this shows that the distribution of the

height at $x = 0$ converges at large time for boundary conditions such that $\epsilon \to \infty$ (e.g. for any fixed $A > -1/2$) to the GSE Tracy-Widom distribution

$$\lim_{t \to \infty} \mathbb{P}(\frac{h(0, t) + \frac{t}{12}}{t^{1/3}} \leqslant s) = F_4(s). \tag{70}$$

## 4.2 Alternative expression for the large-time transition matrix-valued kernel

In this section we will present an equivalent form of the large time transition kernel for the KPZ equation, better suited to study general $\epsilon$. In particular, we will compare it with the transition kernel obtained in in Ref. [37] from the solution of discrete models, i.e. last passage percolation and facilitated TASEP.

To study the large time behavior for general $\epsilon$, i.e. the critical region, it is useful to return to the $\alpha$-decomposition of Section 3.4. Let us choose, as discussed there, $\alpha = A + \frac{1}{2}$. Inserting the replacement (59) into the finite time kernel (7) and repeating the same steps one obtains a large time kernel equivalent to (64) (which, for simplicity, we will also denote by $K^\epsilon$). In that limit the replacement only amounts to multiply by $(w - \epsilon)(z - \epsilon)/(wz)$ the integrand in $K_{22}^\epsilon$, and by $(z - \epsilon)/z$ the integrand in $K_{12}^\epsilon$, leading to

$$K_{11}^\epsilon(r, r') = \frac{1}{4} \iint_{C^2} \frac{dw}{2i\pi} \frac{dz}{2i\pi} \frac{w - z}{w + z} \frac{1}{wz} \frac{\epsilon + w}{w - \epsilon} \frac{\epsilon + z}{z - \epsilon} e^{-rw - r'z + \frac{w^3 + z^3}{3}},$$

$$K_{22}^\epsilon(r, r') = \frac{1}{4} \iint_{C^2} \frac{dw}{2i\pi} \frac{dz}{2i\pi} \frac{w - z}{w + z} \frac{(z + \epsilon)(w + \epsilon)}{wz} e^{-rw - r'z + \frac{w^3 + z^3}{3}}, \tag{71}$$

$$K_{12}^\epsilon(r, r') = \frac{1}{4} \iint_{C^2} \frac{dw}{2i\pi} \frac{dz}{2i\pi} \frac{w - z}{w + z} \frac{z + \epsilon}{wz} \frac{\epsilon + w}{w - \epsilon} e^{-rw - r'z + \frac{w^3 + z^3}{3}}.$$

As advertized above, this choice of $\alpha = A + \frac{1}{2}$ has decreased the number of poles for $w$ or $z$ equal to $\epsilon$, which will be useful below.

We now slightly rewrite this kernel to make more apparent its asymptotic behavior for large arguments $r, r' \to +\infty$. For this purpose it is useful to move the contours of integration. The contour $C$ crosses the real axis between 0 and $\epsilon$. We now choose to move all the contours to the right of $\epsilon$. This will pick some residues that we will evaluate. Doing so, and introducing a contour $\hat{C}$ parallel to the imaginary axis and crossing the real axis at the right of $\epsilon$, we find the decomposition

$$K_{11}^\epsilon(r, r') = \frac{1}{4} \iint_{\hat{C}^2} \frac{dw}{2i\pi} \frac{dz}{2i\pi} \frac{w - z}{w + z} \frac{1}{wz} \frac{\epsilon + w}{w - \epsilon} \frac{\epsilon + z}{z - \epsilon} e^{-rw - r'z + \frac{w^3 + z^3}{3}}$$
$$+ \frac{1}{2} \int_{\hat{C}} \frac{dz}{2i\pi z} e^{\frac{z^3 + \epsilon^3}{3}} \left[ e^{-r'z - r\epsilon} - e^{-rz - r'\epsilon} \right]$$

$$K_{22}^\epsilon(r, r') = \frac{1}{4} \iint_{\hat{C}^2} \frac{dw}{2i\pi} \frac{dz}{2i\pi} \frac{w - z}{w + z} \frac{(z + \epsilon)(w + \epsilon)}{wz} e^{-rw - r'z + \frac{w^3 + z^3}{3}} \tag{72}$$

$$K_{12}^\epsilon(r, r') = \frac{1}{4} \iint_{\hat{C}^2} \frac{dw}{2i\pi} \frac{dz}{2i\pi} \frac{w - z}{w + z} \frac{z + \epsilon}{wz} \frac{\epsilon + w}{w - \epsilon} e^{-rw - r'z + \frac{w^3 + z^3}{3}}$$
$$+ \frac{1}{2} \int_{\hat{C}} \frac{dz}{2i\pi} \frac{z - \epsilon}{z} e^{-r'z - r\epsilon + \frac{z^3 + \epsilon^3}{3}}.$$

This form is useful to evaluate the transitional CDF $F^{(\epsilon)}(s)$ at large $s$, i.e. the Fredholm Pfaffian in a trace expansion at large $s$. The Fredholm Pfaffian involves integrals of the variables $r, r'$



on $[s, +\infty[$. Its behavior for large $s$ is thus controled by the region of large $r, r' > s$. In that region, each term in (72) can be evaluated by saddle point: indeed the new contour $\hat{C}$ can be deformed to include the saddle point. Hence the asymptotics is of the standard Airy type, i.e. each cubic exponential in the integrand leads to a decay $\sim \exp(-\frac{2}{3}s^{3/2})$ and the total decay of each term is thus of the form $\sim \exp(-\frac{2}{3}ps^{3/2})$, where the integer $p = 1, 2$ is the number of cubic exponentials (akin to the number of Airy functions). To be systematic one thus labels the terms in (72) in terms of their homogeneity in cubic exponentials (not counting the terms in $\epsilon^3$). We introduce the notation $K^{(n)}$ when the element $K$ comprises $n$ cubic exponentials and hence we write the kernel as

$$
\begin{aligned}
K_{11}^{\epsilon} &= K_{11}^{\epsilon,(2)} + K_{11}^{\epsilon,(1)}, \\
K_{22}^{\epsilon} &= K_{22}^{\epsilon,(2)}, \\
K_{12}^{\epsilon} &= K_{12}^{\epsilon,(2)} + K_{12}^{\epsilon,(1)}.
\end{aligned}
\tag{73}
$$

*Remark* 4.3. $K_{12}^{\epsilon,(1)}$ is a rank 1 operator.

An important property of our result is that our transition kernel $K^{\epsilon}$ in (72) is equivalent to the cross-over kernel[3] $K_{\mathrm{cross}}$ in Ref. [37] Definition 2.9, i.e. we conjecture that for $\epsilon > 0$

$$
\mathrm{Pf}(J - K^{\epsilon})_{\mathbb{L}^2([s,+\infty[)} = \mathrm{Pf}(J - K_{\mathrm{cross}})_{\mathbb{L}^2([s,+\infty[)}.
\tag{74}
$$

Let us first recall that from Ref. [37] Definition 2.9, the cross-over kernel $K_{\mathrm{cross}}$ is the sum of two kernels $I$ and $R$, the latter having a single non-zero component $R_{22}$. In our notations and contour conventions, these kernels read

$$
\begin{aligned}
I_{11}(r, r') &= \iint_{C^2} \frac{\mathrm{d}z}{2i\pi} \frac{\mathrm{d}w}{2i\pi} \frac{w-z}{w+z} \frac{w+\epsilon}{w} \frac{z+\epsilon}{z} e^{-r'z - rw + \frac{w^3+z^3}{3}}, \\
I_{12}(r, r') &= \frac{1}{2} \iint_{C^2} \frac{\mathrm{d}z}{2i\pi} \frac{\mathrm{d}w}{2i\pi} \frac{w-z}{w+z} \frac{w+\epsilon}{w} \frac{1}{\epsilon - z} e^{-r'z - rw + \frac{w^3+z^3}{3}}, \\
I_{22}(r, r') &= \frac{1}{4} \iint_{\hat{C}^2} \frac{\mathrm{d}z}{2i\pi} \frac{\mathrm{d}w}{2i\pi} \frac{w-z}{w+z} \frac{1}{\epsilon - z} \frac{1}{\epsilon - w} e^{-r'z - rw + \frac{w^3+z^3}{3}}, \\
I_{21}(r, r') &= -I_{12}(r', r), \\
R_{22}(r, r') &= -\frac{1}{4} \int_{\hat{C}} \frac{\mathrm{d}z}{2i\pi} \frac{e^{\frac{z^3+\epsilon^3}{3} - rz - r'\epsilon}}{\epsilon + z} + \frac{1}{4} \int_{\hat{C}} \frac{\mathrm{d}z}{2i\pi} \frac{e^{\frac{z^3+\epsilon^3}{3} - r'z - r\epsilon}}{\epsilon + z} - \frac{1}{2} \int_{C} \frac{\mathrm{d}z}{2i\pi} \frac{z e^{z(r'-r)}}{(\epsilon + z)(\epsilon - z)},
\end{aligned}
\tag{75}
$$

where we recall that the contour $C$ is parallel to the imaginary axis and crosses the real axis between $0$ and $\epsilon$ and that the contour $\hat{C}$ is parallel to the imaginary axis and crosses the real axis at the right of $\epsilon$.

*Remark* 4.4. With these definitions the above formula holds for $\epsilon > 0$. The kernel of Ref. [37] is also valid for $\epsilon < 0$, with the following changes. For $\epsilon < 0$, the contours of $I_{11}$ are unchanged, for $I_{12}$ the contour of $z$ crosses the real axis at the left of $\epsilon$, the contour of $w$ crosses the real axis at the right of $0$ such that $w + z > 0$. For $I_{22}$ both contours of $w$ and $z$ cross the real axis at the right of $0$. For $R_{22}$ the contours cross the real axis between the left of $-\epsilon$ and the right of $\epsilon$.

---

[3]This kernel appeared previously in Refs. [75, 76] . We thank G. Barraquand for pointing these references to us.

It is important to note that for $\epsilon > 0$, the first two terms of $R_{22}$ can be absorbed in $I_{22}$ by changing the contour $\hat{C} \to C$ in $I_{22}$. Indeed the first two terms of $R_{22}$ are nothing more than the residues of the poles of $I_{22}$ at $w = \epsilon$ and $z = \epsilon$. Hence, in later uses, we will redefine for convenience $R_{22}$ to be only its last term

$$R_{22}(r, r') = -\frac{1}{2} \int_C \frac{dz}{2i\pi} \frac{z e^{z(r'-r)}}{(\epsilon + z)(\epsilon - z)} = \frac{1}{4} \text{sgn}(r' - r) e^{-|r-r'|\epsilon} \tag{76}$$

and include the first two terms in $I_{22}$. Since only the sum $I_{22} + R_{22}$ matters this is immaterial.

Since it is already proved in Ref. [37] that in the large positive $\epsilon \to +\infty$ limit, their kernel is the GSE kernel, we see that in this limit it is equivalent to ours. For general $\epsilon > 0$ we have expanded the Fredholm Pfaffians on both sides of Eq. (74) in series of their traces, using the counting in cubic exponentials explained above. It is equivalent to expanding both predictions for the CDF for the height in the transition regimes at large $s$. We found perfect matching up to, and including, third order, i.e. $\exp(-2s^{3/2})$. Thus we have a strong case for the conjecture of the full equivalence of the Fredholm Pfaffians. A complete proof of this identity seems for the moment out of reach.

## 4.3 Small $\epsilon$ behavior of the matrix-valued kernel

In this Section, we will take the small $\epsilon$ limit of the two-dimensional kernel. Since the kernel $K_{\text{cross}}$ converges to the Tracy-Widom GOE kernel as $\epsilon \to 0$, see Ref. [37], the limit of $K^\epsilon$ for $\epsilon \to 0$ will provide a new kernel for the Tracy-Widom GOE distribution in accordance to our conjecture in Eq. (74). The starting point is the expression of $K^\epsilon$ in Eq. (72) and we will show that the $\epsilon \to 0$ limit can be taken without any ambiguity within the different elements of the kernel.

Whenever there is an Airy term of the type $e^{-rz - r\epsilon - z^3/3}$, the $\epsilon \to 0$ limit is clear and we can set $\epsilon = 0$ directly. In the case of the different projectors, where there is solely a term of the type $e^{-r\epsilon}$ in Eq. (72), the limit has to be investigated properly because of possible convergence issue when calculating the Fredholm Pfaffian which involves products of the elements of the matrix-valued kernel, see Eq. (146). Hence, with this remark and expressing the projectors in terms of Airy functions, we rewrite the kernel $K^\epsilon$ as

$$
\begin{aligned}
K_{11}^{\epsilon \to 0}(r, r') = {}& \frac{1}{4} \iint_{\hat{C}^2} \frac{dw}{2i\pi} \frac{dz}{2i\pi} \frac{w - z}{w + z} \frac{1}{wz} e^{-rw - r'z + \frac{w^3 + z^3}{3}} \\
& + \frac{e^{-r\epsilon}}{2} \int_0^{+\infty} d\lambda \text{Ai}(\lambda + r') - \frac{e^{-r'\epsilon}}{2} \int_0^{+\infty} d\lambda \text{Ai}(\lambda + r) \\
K_{22}^{\epsilon \to 0}(r, r') = {}& \frac{1}{4} \iint_{\hat{C}^2} \frac{dw}{2i\pi} \frac{dz}{2i\pi} \frac{w - z}{w + z} e^{-rw - r'z + \frac{w^3 + z^3}{3}} \\
K_{12}^{\epsilon \to 0}(r, r') = {}& \frac{1}{4} \iint_{\hat{C}^2} \frac{dw}{2i\pi} \frac{dz}{2i\pi} \frac{w - z}{w + z} \frac{1}{w} e^{-rw - r'z + \frac{w^3 + z^3}{3}} + \frac{e^{-r\epsilon}}{2} \text{Ai}(r') - \frac{\epsilon e^{-r\epsilon}}{2} \int_0^{+\infty} d\lambda \text{Ai}(\lambda + r').
\end{aligned}
\tag{77}
$$

If one defines the following vectors

$$|u_\epsilon(r)\rangle = \begin{pmatrix} \frac{e^{-r\epsilon}}{2} \\ 0 \end{pmatrix}, \qquad |v_\epsilon(r')\rangle = \begin{pmatrix} \int_0^{+\infty} d\lambda \text{Ai}(\lambda + r') \\ \text{Ai}(r') - \epsilon \int_0^{+\infty} d\lambda \text{Ai}(\lambda + r'), \end{pmatrix} \tag{78}$$

then the kernel $K^{\epsilon \to 0}$ admits the reduced representation

$$K^{\epsilon \to 0} = K^{\infty} + |u_\epsilon\rangle \langle v_\epsilon| - |v_\epsilon\rangle \langle u_\epsilon|, \tag{79}$$

where $K^{\infty}$ is the GSE kernel of Eq. (69). In order to treat the $\epsilon$ dependence in the term $|u_\epsilon\rangle \langle v_\epsilon| - |v_\epsilon\rangle \langle u_\epsilon|$, we treat it as a rank one perturbation to a Fredholm Pfaffian.

$$\mathrm{Pf}(J - K^{\epsilon \to 0})_{\mathbb{L}^2([s,+\infty[)} = \mathrm{Pf}(J - K^{\infty} - |u_\epsilon\rangle \langle v_\epsilon| + |v_\epsilon\rangle \langle u_\epsilon|)_{\mathbb{L}^2([s,+\infty[)}. \tag{80}$$

By a generalization of the matrix determinant lemma and the Sherman Morrison formula for Fredholm determinants[4], we have

$$\mathrm{Pf}(J - K^{\infty} - |u_\epsilon\rangle \langle v_\epsilon| + |v_\epsilon\rangle \langle u_\epsilon|) = \mathrm{Pf}(J - K^{\infty})\left[1 - \langle u_\epsilon|(I + JK^{\infty})^{-1}J\,|v_\epsilon\rangle\right]. \tag{81}$$

The remaining goal is to show that the $\epsilon \to 0$ limit can be taken in the inner product in Eq. (81). It is possible to expand the resolvant $(I + JK^{\infty})^{-1}$, we will show on the lowest order term of the expansion that the $\epsilon \to 0$ limit can be taken easily and we will conjecture the same conclusion for higher order terms.

$$
\begin{aligned}
\langle u_\epsilon|J\,|v_\epsilon\rangle - \langle u_0|J\,|v_0\rangle &= \frac{1}{2}\int_s^{+\infty} dr\left[(e^{-r\epsilon} - 1)\mathrm{Ai}(r) - \epsilon e^{-r\epsilon}\int_0^{+\infty} d\lambda\,\mathrm{Ai}(\lambda + r)\right]\\
&= \frac{1}{2}\int_0^{+\infty} dr(e^{-s\epsilon - r\epsilon} - 2 + e^{-r\epsilon})\mathrm{Ai}(r + s)\\
&\underset{\epsilon \to 0}{=} \mathcal{O}(\epsilon).
\end{aligned}
\tag{82}
$$

From the first to the second line, we proceeded to the change of variable $(r, \lambda) \to (u = r + \lambda \in \mathbb{R}_+, v = r - \lambda \in [-u, u])$ and relabeled $u \to r$.

Our conclusion is that we can simply take the $\epsilon \to 0$ limit from the beginning in the vectors $|u_\epsilon\rangle$ and $|v_\epsilon\rangle$ so that the final resulting infinite-time kernel for $A = -1/2$ reads

$$
\begin{aligned}
K_{11}^0(r, r') &= \frac{1}{4}\iint_{\hat{C}^2} \frac{dw}{2i\pi}\frac{dz}{2i\pi}\frac{w - z}{w + z}\frac{1}{wz}e^{-rw - r'z + \frac{w^3 + z^3}{3}}\\
&\qquad\qquad + \frac{1}{2}\int_0^{+\infty} d\lambda\,\mathrm{Ai}(\lambda + r') - \frac{1}{2}\int_0^{+\infty} d\lambda\,\mathrm{Ai}(\lambda + r)\\
K_{22}^0(r, r') &= \frac{1}{4}\iint_{\hat{C}^2} \frac{dw}{2i\pi}\frac{dz}{2i\pi}\frac{w - z}{w + z}e^{-rw - r'z + \frac{w^3 + z^3}{3}}\\
K_{12}^0(r, r') &= \frac{1}{4}\iint_{\hat{C}^2} \frac{dw}{2i\pi}\frac{dz}{2i\pi}\frac{w - z}{w + z}\frac{1}{w}e^{-rw - r'z + \frac{w^3 + z^3}{3}} + \frac{1}{2}\mathrm{Ai}(r').
\end{aligned}
\tag{83}
$$

*Remark* 4.5. Note that in the kernel $K^0$, the derivative relation between its elements is conserved: $K_{22}^0(r, r') = \partial_r \partial_{r'} K_{11}^0(r, r')$, $K_{12}^0(r, r') = -\partial_{r'} K_{11}^0(r, r')$ and $K_{22}^0(r, r') = -\partial_r K_{12}^0(r, r')$.

According to Ref. [37] and our conjecture in Eq. (74), the Fredholm Pfaffian of the kernel $K^0$ describes the Tracy-Widom GOE distribution. To the best of our knowledge, this kernel $K^0$ has not appeared before in the literature and therefore provides an alternative expression for the Tracy-Widom GOE distribution. Since it enjoys the same simple derivative structure as the GSE kernel, see Remark 4.5 above, without involving any sign function, contrary to $K_{\mathrm{cross}}$, we hope this kernel will find further applications.

---

[4]We obtain this generalization by considering the identity $\mathrm{Pf}(J - K)^2 = \mathrm{Det}(I + JK)$ and treat the term $J(|u_\epsilon\rangle \langle v_\epsilon| - |v_\epsilon\rangle \langle u_\epsilon|)$ as a rank two perturbation.

### 4.4 Solution for finite time at $A = -1/2$

Having understood the $\epsilon \to 0$ limit on the infinite time matrix-valued kernel, we extend the calculation to study the critical case $A = -1/2$ at finite time. To this aim, we first rewrite the kernel Eq. (7) using the $\alpha$-prescription with $\alpha = A + \frac{1}{2}$.

$$
\begin{aligned}
K_{11}(r, r') &= \iint_{C^2} \frac{dw}{2i\pi} \frac{dz}{2i\pi} \frac{w-z}{w+z} \frac{\Gamma(A+\frac{1}{2}-w)}{\Gamma(A+\frac{1}{2}+w)} \frac{\Gamma(A+\frac{1}{2}-z)}{\Gamma(A+\frac{1}{2}+z)} \\
&\quad \times \Gamma(2w)\Gamma(2z)\cos(\pi(w-A-\frac{1}{2}))\cos(\pi(z-A-\frac{1}{2}))e^{-rw-r'z+t\frac{w^3+z^3}{3}}, \\
K_{22}(r, r') &= \iint_{C^2} \frac{dw}{2i\pi} \frac{dz}{2i\pi} \frac{w-z}{w+z} \frac{\Gamma(A+\frac{1}{2}-w)}{\Gamma(A+\frac{1}{2}+w)} \frac{\Gamma(A+\frac{1}{2}-z)}{\Gamma(A+\frac{1}{2}+z)} \\
&\quad \times \Gamma(2w)\Gamma(2z) \frac{\sin(\pi(w-A-\frac{1}{2}))}{\pi} \frac{\sin(\pi(z-A-\frac{1}{2}))}{\pi} e^{-rw-r'z+t\frac{w^3+z^3}{3}}, \quad (84)\\
K_{12}(r, r') &= \iint_{C^2} \frac{dw}{2i\pi} \frac{dz}{2i\pi} \frac{w-z}{w+z} \frac{\Gamma(A+\frac{1}{2}-w)}{\Gamma(A+\frac{1}{2}+w)} \frac{\Gamma(A+\frac{1}{2}-z)}{\Gamma(A+\frac{1}{2}+z)} \\
&\quad \times \Gamma(2w)\Gamma(2z)\cos(\pi(w-A-\frac{1}{2})) \frac{\sin(\pi(z-A-\frac{1}{2}))}{\pi} e^{-rw-r'z+t\frac{w^3+z^3}{3}}, \\
K_{21}(r, r') &= -K_{12}(r', r).
\end{aligned}
$$

We recall that the contours $C$ are parallel to the imaginary axis and cross the real axis between 0 and $A + \frac{1}{2}$. Let us now push this contour to the right of $A + \frac{1}{2}$ by picking the related pole and call the new contour $\hat{C}$. We will directly write the result at $A = -1/2$ without further details to conclude this Section on the matrix-valued kernel.

$$
\begin{aligned}
K_{11}(r, r') &= \iint_{\hat{C}^2} \frac{dw}{2i\pi} \frac{dz}{2i\pi} \frac{w-z}{w+z} \frac{\Gamma(-w)}{\Gamma(w)} \frac{\Gamma(-z)}{\Gamma(z)} \Gamma(2w)\Gamma(2z)\cos(\pi w)\cos(\pi z)e^{-rw-r'z+t\frac{w^3+z^3}{3}} \\
&\quad + \int_{\hat{C}} \frac{dz}{2i\pi} \frac{\Gamma(1-z)}{\Gamma(1+z)} \Gamma(2z)\cos(\pi z)e^{t\frac{z^3}{3}} \left[ e^{-r'z} - e^{-rz} \right], \\
K_{22}(r, r') &= \iint_{\hat{C}^2} \frac{dw}{2i\pi} \frac{dz}{2i\pi} \frac{w-z}{w+z} \frac{\Gamma(-w)}{\Gamma(w)} \frac{\Gamma(-z)}{\Gamma(z)} \Gamma(2w)\Gamma(2z) \frac{\sin(\pi w)}{\pi} \frac{\sin(\pi z)}{\pi} e^{-rw-r'z+t\frac{w^3+z^3}{3}}, \\
K_{12}(r, r') &= \iint_{\hat{C}^2} \frac{dw}{2i\pi} \frac{dz}{2i\pi} \frac{w-z}{w+z} \frac{\Gamma(-w)}{\Gamma(w)} \frac{\Gamma(-z)}{\Gamma(z)} \Gamma(2w)\Gamma(2z)\cos(\pi w) \frac{\sin(\pi z)}{\pi} e^{-rw-r'z+t\frac{w^3+z^3}{3}} \\
&\quad + \int_{\hat{C}} \frac{dz}{2i\pi} \frac{\Gamma(1-z)}{\Gamma(1+z)} \Gamma(2z) \frac{\sin(\pi z)}{\pi} e^{-r'z+t\frac{z^3}{3}}, \\
K_{21}(r, r') &= -K_{12}(r', r).
\end{aligned}
$$

$$(85)$$

This final kernel is the one that describes the critical case $A = -1/2$ at all times.

It was shown in Ref. [27] that Fredholm Pfaffians with matrix-valued kernel of Schur type can also be expressed as Fredholm determinants with a scalar kernel. We now dedicate the rest of this work to the understanding of the structure of the scalar-valued kernel.

# 5 From a matrix valued kernel to a scalar kernel

## 5.1 Solution for the KPZ generating function at all times for generic $A$ in terms of a scalar kernel

The general kernel we have obtained in Eq. (7) has a particular structure in the form of a Schur Pfaffian. With this structure, the kernel verifies the hypothesis of Proposition B.2 of [27]. This proposition states that we can transform the Fredholm Pfaffian of Eq. (54) which involves a matrix valued kernel, into a Fredholm determinant of a scalar kernel. To proceed, let us first define the functions

$$
\begin{aligned}
f_{\text{odd}}(r) &= \int_C \frac{dw}{2i\pi} \frac{\Gamma(A + \frac{1}{2} - w)}{\Gamma(A + \frac{1}{2} + w)} \Gamma(2w) \cos(\pi w) e^{-rw + t\frac{w^3}{3}} \\
f_{\text{even}}(r) &= \int_C \frac{dz}{2i\pi} \frac{\Gamma(A + \frac{1}{2} - z)}{\Gamma(A + \frac{1}{2} + z)} \Gamma(2z) \frac{\sin(\pi z)}{\pi} e^{-rz + t\frac{z^3}{3}},
\end{aligned}
\tag{86}
$$

and the kernel $\bar{K}_{t,\varsigma}$ such that for all $(x, y) \in \mathbb{R}_+^2$

$$
\begin{aligned}
\bar{K}_{t,\varsigma}(x, y) &= 2\partial_x \int_{\mathbb{R}} dr \frac{\varsigma}{\varsigma + e^{-r}} [f_{\text{even}}(r + x) f_{\text{odd}}(r + y) - f_{\text{odd}}(r + x) f_{\text{even}}(r + y)] \\
&= 2\partial_x \int_{\mathbb{R}} dr \frac{\varsigma}{\varsigma + e^{-r}} \iint_{C^2} \frac{dw dz}{(2i\pi)^2} \frac{\Gamma(A + \frac{1}{2} - w)}{\Gamma(A + \frac{1}{2} + w)} \frac{\Gamma(A + \frac{1}{2} - z)}{\Gamma(A + \frac{1}{2} + z)} \\
&\quad \times \Gamma(2w) \Gamma(2z) \frac{\sin(\pi(z - w))}{\pi} e^{-xz - yw - rw - rz + t\frac{w^3 + z^3}{3}}.
\end{aligned}
\tag{87}
$$

We recall that in this formula the contours $C$ are parallel to the imaginary axis and cross the real axis between 0 and $A + \frac{1}{2}$.

Then, the Laplace transform of the one-point distribution of the exponential of the KPZ height admits the following representation:

$$
g(\varsigma) = \mathbb{E}_{\text{KPZ}} \left[ \exp(-\varsigma e^{H(t)}) \right] = \text{Pf}(J - \sigma_\varsigma K)_{\mathbb{L}^2(\mathbb{R})} = \sqrt{\text{Det}(I - \bar{K}_{t,\varsigma})_{\mathbb{L}^2(\mathbb{R}_+)}}.
\tag{88}
$$

*Remark* 5.1. In the case of the scalar kernel, there is no $\alpha$-decomposition as any $\alpha$ prescription in the functions $f_{\text{odd}}$ and $f_{\text{even}}$ will eventually lead to the same kernel (87).

*Large time limit.* To obtain the large time limit of (88) one performs the same rescaling as in Sec. 4, namely one chooses $\varsigma = e^{-t^{1/3}s}$ and one rescales $(w, z) \to t^{-1/3}(w, z)$, $r \to t^{1/3}r$. The Fermi factor $\sigma_\varsigma$ produces a Heaviside function $\Theta(r - s)$ and then (88) becomes (11), in terms of the large time scalar kernel $\bar{K}$ given in Eq. (12).

## 5.2 Large $\epsilon$ behavior of the scalar kernel and convergence to the GSE

We recall that at large time the critical cross-over parameter is given by $\epsilon = (A + \frac{1}{2})t^{1/3}$. For large $\epsilon$, we use the exact same procedure as in Section 4.1 and we obtain from Eq. (12) the kernel

$$
\bar{K}^{\infty}(x, y) = \frac{1}{2} \iint_{C^2} \frac{dw dz}{(2i\pi)^2} \frac{w - z}{w + z} \frac{1}{w} e^{-xz - yw + \frac{w^3 + z^3}{3}} := K^{\text{GLD}}(x, y),
\tag{89}
$$

which was proved to be the one-dimensional GSE kernel in Refs. [24, 27]. Performing the integrals one can equivalently write it as

$$K^{\text{GLD}}(x, y) = K_{\text{Ai}}(x, y) - \frac{1}{2}\text{Ai}(x) \int_0^{+\infty} d\lambda \text{Ai}(y + \lambda), \tag{90}$$

where $K_{\text{Ai}}$ is the standard Airy kernel. This is the form in which it was obtained in [24] for the case $A = +\infty$. Here we see that the GSE-TW holds for any fixed $A > -1/2$ at large time, consistent with the conclusion obtained above from the matrix valued kernel.

## 5.3 Small $\epsilon$ limit of the scalar kernel and convergence to the GOE

To investigate the small $\epsilon$ behavior of the scalar kernel, one proceeds as in the matrix case by moving the contours $C$ of the scalar kernel of Eq. (12) to $\hat{C}$ which is to the right of $\epsilon$, and collecting all residues along the way. Doing so, we rewrite the scalar kernel as

$$\bar{K}(x, y) = \frac{1}{2} \iint_{\hat{C}^2} \frac{dw dz}{(2i\pi)^2} \frac{\epsilon + w}{\epsilon - w} \frac{\epsilon + z}{\epsilon - z} \frac{w - z}{w + z} \frac{1}{w} e^{-xz - yw + \frac{w^3 + z^3}{3}}$$
$$- \epsilon e^{-\epsilon x + \frac{\epsilon^3}{3}} \int_0^{+\infty} d\lambda \text{Ai}(y + \lambda) + \text{Ai}(x) e^{-\epsilon y + \frac{\epsilon^3}{3}}. \tag{91}$$

The new contour $\hat{C}$ is now well suited for the small $\epsilon$ limit in the first term. However one cannot simply set $\epsilon = 0$ in the last two terms. Indeed the operator product of the last term with the second one leads to a divergent integral. We now study this delicate limit in details.

To further study the small $\epsilon$ limit, one first conjugates the kernel on the left by the multiplication of $e^{-\epsilon x}$ and on the right by the multiplication of $e^{\epsilon y}$. This operation leaves the associated Fredholm determinant unchanged. The new kernel obtained by this manipulation is the following

$$\mathsf{K}(x, y) = \frac{1}{2} \iint_{\hat{C}^2} \frac{dw dz}{(2i\pi)^2} \frac{\epsilon + w}{\epsilon - w} \frac{\epsilon + z}{\epsilon - z} \frac{w - z}{w + z} \frac{1}{w} e^{-xz - yw + \epsilon y - \epsilon x + \frac{w^3 + z^3}{3}}$$
$$- \epsilon e^{-2\epsilon x + \epsilon y + \frac{\epsilon^3}{3}} \int_0^{+\infty} d\lambda \text{Ai}(y + \lambda) + \text{Ai}(x) e^{-\epsilon x + \frac{\epsilon^3}{3}}. \tag{92}$$

We split this kernel into three components and we will analyze them separately.

$$\mathsf{K}_0(x, y) = \frac{1}{2} \iint_{\hat{C}^2} \frac{dw dz}{(2i\pi)^2} \frac{\epsilon + w}{\epsilon - w} \frac{\epsilon + z}{\epsilon - z} \frac{w - z}{w + z} \frac{1}{w} e^{-xz - yw + \epsilon y - \epsilon x + \frac{w^3 + z^3}{3}},$$
$$\mathsf{K}_1(x, y) = \epsilon e^{-2\epsilon x + \epsilon y + \frac{\epsilon^3}{3}} \int_0^{+\infty} d\lambda \text{Ai}(y + \lambda), \tag{93}$$
$$\mathsf{K}_2(x, y) = \text{Ai}(x) e^{-\epsilon x + \frac{\epsilon^3}{3}}.$$

- The small $\epsilon$ limit of $\mathsf{K}_0$ is taken easily as $\epsilon$ does not play any particular role in the convergence of the integrals, it reads

$$\lim_{\epsilon \to 0} \mathsf{K}_0 = K^{\text{GLD}}. \tag{94}$$

- Similarly, the small $\epsilon$ limit of $\mathsf{K}_2$ is taken straightforwardly and reads

$$\lim_{\epsilon \to 0} \mathsf{K}_2 = \text{Ai}. \tag{95}$$

Indeed, the Airy function will always make any integral over $x$ convergent.

- The convergence of $\mathsf{K}_1$ is more complicated and as the devil hides in the details, we have to analyze this part of the kernel very carefully. One could be tempted to write that

$$\lim_{\epsilon \to 0} \mathsf{K}_1(x, y) = \lim_{\epsilon \to 0} \epsilon e^{-2\epsilon x + \epsilon y + \frac{\epsilon^3}{3}} \int_0^{+\infty} d\lambda \mathrm{Ai}(y + \lambda) \underset{?}{=} 0. \tag{96}$$

This is wrong and the whole subtlety of the small $\epsilon$ limit. The convergence of the kernel with respect to the variable $y$ is controlled by the Airy function, hence we are allowed to take the limit $e^{\epsilon y + \frac{\epsilon^3}{3}} \to 1$. We are now going to show the distributional identity

$$\lim_{\epsilon \to 0} \left( \epsilon e^{-2\epsilon x} \right) = \frac{1}{2} \delta(x = +\infty) := \frac{1}{2} \delta_\infty(x). \tag{97}$$

When we multiply the kernel $\mathsf{K}_1$ on the left by an arbitrary element $K$, we have to calculate the following product

$$\int_s^{+\infty} dx K(z, x) \mathsf{K}_1(x, y) = \epsilon \int_s^{+\infty} dx K(z, x) e^{-2\epsilon x} \int_0^{+\infty} d\lambda \mathrm{Ai}(y + \lambda). \tag{98}$$

Proceeding to the change of variable $\tilde{x} = 2\epsilon x$, we have

$$\int_s^{+\infty} dx K(z, x) \mathsf{K}_1(x, y) = \frac{1}{2} \int_{2s\epsilon}^{+\infty} d\tilde{x} K(z, \frac{\tilde{x}}{2\epsilon}) e^{-\tilde{x}} \int_0^{+\infty} d\lambda \mathrm{Ai}(y + \lambda). \tag{99}$$

In the limit of small $\epsilon$, this integral should be equal to

$$\frac{1}{2} \lim_{\epsilon \to 0} \int_{2s\epsilon}^{+\infty} d\tilde{x} K(z, \frac{\tilde{x}}{2\epsilon}) e^{-\tilde{x}} \int_0^{+\infty} d\lambda \mathrm{Ai}(y + \lambda)$$

$$= \frac{1}{2} \lim_{X \to +\infty} K(z, X) \left( \int_0^{+\infty} d\tilde{x} e^{-\tilde{x}} \right) \int_0^{+\infty} d\lambda \mathrm{Ai}(y + \lambda) \tag{100}$$

$$= \frac{1}{2} \lim_{X \to +\infty} K(z, X) \int_0^{+\infty} d\lambda \mathrm{Ai}(y + \lambda).$$

Naively one would claim that this limit is always zero due to some exponential decay of the kernels. However, the kernel $\mathsf{K}_2$ does not decay

$$\lim_{X \to +\infty} \mathsf{K}_2(z, X) = \mathrm{Ai}(z) \neq 0, \tag{101}$$

hence products of $\mathsf{K}_2$ and $\mathsf{K}_1$ give a non zero result for $\epsilon \to 0$.

To summarize this discussion, we have shown that the kernel for $\epsilon = 0$ is equal to

$$\mathsf{K}(x, y) = K^{\mathrm{GLD}}(x, y) - \frac{1}{2} \delta_\infty(x) \int_0^{+\infty} d\lambda \mathrm{Ai}(y + \lambda) + \mathrm{Ai}(x). \tag{102}$$

The last effort of this Section now consists in proving that the Fredholm determinant of $\mathsf{K}$ is indeed identical to the one of the GOE-TW distribution. As it is common for such manipulations, we introduce the brackets notations

$$\delta_\infty(x) = |\delta_\infty\rangle$$

$$\int_0^{+\infty} d\lambda \, \mathrm{Ai}(y + \lambda) = \langle 1 | \mathrm{Ai} \tag{103}$$

$$\mathrm{Ai}(x) = |\mathrm{Ai}\rangle,$$

where we have implicitly promoted the Airy function to an operator of kernel $\mathrm{Ai}(x, y) = \mathrm{Ai}(x + y)$.

*Remark* 5.2. With this notation, we have $K^{\text{GLD}} = K_{\text{Ai}} - \frac{1}{2} |\text{Ai}\rangle \langle 1| \text{Ai}$.

Under these conventions, our kernel for $\epsilon = 0$ is written as

$$\mathsf{K} = K^{\text{GLD}} - \frac{1}{2} |\delta_\infty\rangle \langle 1| \text{Ai} + |\text{Ai}\rangle \langle 1|. \tag{104}$$

The kernel $\mathsf{K}$ can be interpreted as the $K^{\text{GLD}}$ kernel with a rank 2 perturbation. By the matrix determinant lemma coupled to the Sherman-Morrison formula, we can evaluate the Fredholm determinant of $\mathsf{K}$ and extract the two rank 1 perturbations as

$$\begin{aligned}
\text{Det}(I - \mathsf{K}) = \text{Det}\left(I - K^{\text{GLD}} + \frac{1}{2} |\delta_\infty\rangle \langle 1| \text{Ai} - |\text{Ai}\rangle \langle 1|\right) &= \text{Det}\left(I - K^{\text{GLD}}\right) \\
\times \bigg[ \left(1 - \langle 1| (I - K^{\text{GLD}})^{-1} |\text{Ai}\rangle\right) &\left(1 + \frac{1}{2} \langle 1| \text{Ai}(I - K^{\text{GLD}})^{-1} |\delta_\infty\rangle\right) \\
&+ \frac{1}{2} \langle 1| \text{Ai}(I - K^{\text{GLD}})^{-1} |\text{Ai}\rangle \langle 1| (I - K^{\text{GLD}})^{-1} |\delta_\infty\rangle \bigg].
\end{aligned} \tag{105}$$

Since $K^{\text{GLD}}$ decreases fast enough at infinity in both arguments, we have for any integer $n \geqslant 1$ the equality

$$(K^{\text{GLD}})^n |\delta_\infty\rangle = 0. \tag{106}$$

On the contrary, the identity keeps $|\delta_\infty\rangle$ invariant, i.e. $I |\delta_\infty\rangle = |\delta_\infty\rangle$ and hence by a regular series expansion we obtain the action of $\delta_\infty$ on the resolvant of $K^{\text{GLD}}$.

$$(I - K^{\text{GLD}})^{-1} |\delta_\infty\rangle = |\delta_\infty\rangle. \tag{107}$$

Similarly to the above considerations, by the decay property of the Airy function, we have that $\langle 1| \text{Ai}|\delta_\infty\rangle = 0$ and we also have readily the inner product $\langle 1|\delta_\infty\rangle = 1$. Implementing these results in the Fredholm determinant of Eq. (105) leads to

$$\begin{aligned}
\text{Det}(I - \mathsf{K}) &= \text{Det}\left(I - K^{\text{GLD}}\right) \left[1 - \langle 1| (1 - \frac{1}{2}\text{Ai})(I - K^{\text{GLD}})^{-1} |\text{Ai}\rangle\right] \\
&= \text{Det}\left(I - K^{\text{GLD}} - |\text{Ai}\rangle \langle 1| (1 - \frac{1}{2}\text{Ai})\right) \\
&= \text{Det}\left(I - K_{\text{Ai}} - |\text{Ai}\rangle \langle 1| (1 - \text{Ai})\right).
\end{aligned} \tag{108}$$

In the first line, we simplified Eq. (105) using the action of $\delta_\infty$, from the first to the second line we used the Matrix determinant lemma and from the second to the third line we introduced the expression of $K^{\text{GLD}}$ with brackets notations. The last Fredholm determinant was shown by Forrester in Ref. [77] to be related to the GOE Tracy-Widom cumulative distribution function $F_1(s)$ as

$$F_1(s)^2 = \text{Det}\left(I - K_{\text{Ai}} - |\text{Ai}\rangle \langle 1| (1 - \text{Ai})\right)_{\mathbb{L}^2([s, +\infty[)}. \tag{109}$$

Hence, we finally conclude that for $\epsilon \to 0^+$, the cumulative probability of the one-point KPZ height field is given by the GOE Tracy-Widom distribution.

$$\lim_{t \to \infty} \mathbb{P}\left(\frac{H(t)}{t^{1/3}} \leqslant s\right) = F_1(s). \tag{110}$$

## 5.4 Solution for finite time at $A = -1/2$

Now that we have understood how to handle the limit $\epsilon \to 0^+$ on the infinite time kernel we can extend the same method to study $A = -1/2$ at finite time. We first rewrite the finite time scalar kernel (87) for any $A > -1/2$ as follows

$$
\bar{K}_{t,\varsigma}(x,y) = -\iint_{C^2} \frac{\mathrm{d}w\mathrm{d}z}{(2i\pi)^2} \frac{\Gamma(1+A+\frac{1}{2}-w)}{\Gamma(1+A+\frac{1}{2}+w)} \frac{\Gamma(1+A+\frac{1}{2}-z)}{\Gamma(1+A+\frac{1}{2}+z)} \frac{A+\frac{1}{2}+w}{A+\frac{1}{2}-w} \frac{A+\frac{1}{2}+z}{A+\frac{1}{2}-z}
$$
$$
\times \Gamma(2w)\Gamma(1+2z)\varsigma^{z+w} \frac{\sin(\pi(z-w))}{\sin(\pi(z+w))} e^{-xz-yw+t\frac{w^3+z^3}{3}}, \tag{111}
$$

where we have performed the derivative $\partial_x$, rearranged the Gamma functions and used the alternative form of the Mellin Barnes formula (46). We consider now the case $0 < A + \frac{1}{2} < 1$ (since we will perform the limit $A \to -1/2$ below this is sufficient for our purpose). Then we can move the integration contour of both variables from $C$ to $\hat{C}$ which, at finite time is $\hat{C} = a + i\mathbb{R}$ with $a \in ]A+\frac{1}{2}, 1[$. This allows to take the $A = -1/2$ limit, following similar steps as in the previous Section.

To study the $A \to -\frac{1}{2}$ limit, we will first denote $\tilde{A} = A + \frac{1}{2}$ and similarly to the infinite time limit, we conjugate the kernel $\bar{K}_{t,\varsigma}$ on the left by the multiplication of $e^{-\tilde{A}x}$ and on the right by the multiplication of $e^{\tilde{A}y}$ and we denote $\mathsf{K}_{t,\varsigma}$ the new kernel. This operation leaves again the associated Fredholm determinant unchanged. Evaluating the two residues of the kernel splits it in three parts $\mathsf{K}_{t,\varsigma} = K_0^{\tilde{A}} + K_1^{\tilde{A}} + K_2^{\tilde{A}}$. The first part is the same expression as (111) with $C$ replaced by $\hat{C}$. We will write it here with the limit $A \to -1/2$ already taken

$$
K_0^0(x,y)
$$
$$
= \frac{\partial_x}{2\pi} \iint_{\hat{C}^2} \frac{\mathrm{d}w\mathrm{d}z}{(2i\pi)^2} \Gamma(-w)\Gamma(-z)\Gamma(w+\frac{1}{2})\Gamma(z+\frac{1}{2})(4\varsigma)^{z+w} \frac{\sin(\pi(z-w))}{\sin(\pi(z+w))} e^{-xz-yw+t\frac{w^3+z^3}{3}}. \tag{112}
$$

We now determine the residues at $w, z = \tilde{A}$ obtained by pushing the contour $C$ into $\hat{C}$.

- The residue at $z = \tilde{A}$ gives the projector

$$
K_1^{\tilde{A}}(x,y) = -\varsigma^{\tilde{A}} \tilde{A} e^{\tilde{A}y - 2x\tilde{A} + t\frac{\tilde{A}^3}{3}} \psi_{\tilde{A}}(y), \tag{113}
$$

  with

$$
\psi_{\tilde{A}}(y) = -2 \int_{\hat{C}} \frac{\mathrm{d}w}{2i\pi} \frac{\Gamma(1+\tilde{A}-w)}{\Gamma(1+\tilde{A}+w)} \frac{\tilde{A}+w}{\tilde{A}-w} \Gamma(2w)\varsigma^w \frac{\sin(\pi(w-\tilde{A}))}{\sin(\pi(w+\tilde{A}))} e^{-yw+t\frac{w^3}{3}}. \tag{114}
$$

  We only need the $\tilde{A} = 0$ limit of $\psi_{\tilde{A}}(y)$ given as

$$
\psi_0(y) = -\frac{1}{\sqrt{\pi}} \int_{\hat{C}} \frac{\mathrm{d}w}{2i\pi} \Gamma(-w)\Gamma(w+\frac{1}{2})(4\varsigma)^w e^{-yw+t\frac{w^3}{3}}, \tag{115}
$$

  where in that limit $\hat{C}$ crosses the real axis in the interval $]0,1[$. As in the infinite time limit, the kernel $K_1^{\tilde{A}}$ converges to

$$
\lim_{\tilde{A}\to 0} K_1^{\tilde{A}}(x,y) = -\frac{1}{2}\delta_\infty(x)\psi_0(y). \tag{116}
$$

*Remark* 5.3. Introducing the Fourier transform of the Airy function and the Barnes integral

$$e^{t\frac{w^3}{3}} = \int_{\mathbb{R}} dr\, \mathrm{Ai}(r) e^{t^{1/3}wr} \,,$$

$$_2F_1(a,b,c,z) = \frac{\Gamma(c)}{\Gamma(a)\Gamma(b)} \int_{i\mathbb{R}} \frac{ds}{2i\pi}(-z)^s \frac{\Gamma(a+s)\Gamma(b+s)\Gamma(-s)}{\Gamma(c+s)} \,,$$

(117)

it is possible to express $\psi_0(y)$ as

$$\psi_0(y) = \int_{\mathbb{R}} dr\, \mathrm{Ai}(r) \left[ 1 - \frac{1}{\sqrt{1+4\varsigma e^{-y+t^{1/3}r}}} \right].$$

(118)

This is reminiscent of the result obtained in Refs. [26,58] about the solution at all times for $A = -1/2$, see below.

- The residue at $w = \tilde{A}$ gives another projector

$$K_2^{\tilde{A}}(x,y) = -\varsigma^{\tilde{A}} e^{-\tilde{A}x + t\frac{\tilde{A}^3}{3}} \partial_x \psi_{\tilde{A}}(x).$$

(119)

In the limit $A \to -\frac{1}{2}$, this kernel converges to

$$\lim_{\tilde{A}\to 0} K_2^{\tilde{A}}(x,y) = \phi_0(x) := -\partial_x \psi_0(x) \,, \quad \psi_0(x) = \int_0^{+\infty} d\lambda\, \phi_0(x+\lambda).$$

(120)

To summarize, the $A \to -\frac{1}{2}$ limit of the kernel $\mathsf{K}_{t,\varsigma}$ is given by

$$\lim_{\tilde{A}\to 0} \mathsf{K}_{t,\varsigma}(x,y) = K_0^0(x,y) - \frac{1}{2}\delta_\infty(x)\psi_0(y) - \partial_x \psi_0(x)$$

$$= K_0^0(x,y) - \frac{1}{2}\delta_\infty(x) \int_0^{+\infty} d\lambda\, \phi_0(y+\lambda) + \phi_0(x).$$

(121)

The structure of this kernel is similar to the one for infinite time in (104). Hence we can follow the same steps and treat the two projectors and the $\delta_\infty$ term using the Sherman-Morrison formula and the matrix determinant lemma which allows us to conclude that

$$\lim_{\tilde{A}\to 0} \mathrm{Det}\left(I - \bar{K}_{t,\varsigma}\right) = \mathrm{Det}\left(I - K_0^0 - |\phi_0\rangle\langle 1|(1-\frac{1}{2}\phi_0)\right)_{\mathbb{L}^2(\mathbb{R}_+)},$$

(122)

which translates for the generating function of the exponential of the KPZ height as

$$g(\varsigma)^2 = \mathbb{E}_{\mathrm{KPZ}}\left[\exp(-\varsigma e^{H(t)})\right]^2 = \mathrm{Det}\left(I - K_0^0 - |\phi_0\rangle\langle 1|(1-\frac{1}{2}\phi_0)\right)_{\mathbb{L}^2(\mathbb{R}_+)}.$$

(123)

Our final result for the generating function at all time for $A = -1/2$ can thus be written more explicitly as

$$g(\varsigma) = \mathbb{E}_{\mathrm{KPZ}}\left[\exp(-\varsigma e^{H(t)})\right] = \sqrt{\mathrm{Det}\left(I - K_\varsigma\right)_{\mathbb{L}^2(\mathbb{R}_+)}},$$

(124)

with

$$K_\varsigma(x,y) = K_0^0(x,y) - \frac{1}{2}\phi_0(x) \int_0^{+\infty} d\lambda\, \phi_0(y+\lambda) + \phi_0(x),$$

(125)

where $K_0^0(x, y)$ is given in (112) and

$$\phi_0(x) = \frac{1}{\sqrt{\pi}} \int_{\hat{C}} \frac{dw}{2i\pi} \Gamma(1-w)\Gamma\left(w + \frac{1}{2}\right)(4\varsigma)^w e^{-xw + t\frac{w^3}{3}}, \tag{126}$$

where in all formula the contour $\hat{C}$ runs parallel to the imaginary axis and crosses the real axis in the interval $]0, 1[$. In the large time limit one sets $\varsigma = e^{-t^{1/3}s}$ and rescales the integration variables $w \to wt^{-1/3}, z \to zt^{-1/3}$ and perform a similarity transformation on the kernel. The function $\phi_0$ then leads to the Airy function, and one recovers the large time result of the previous Section.

*Remark* 5.4. Note that in Refs. [26,58], another formula was proved for finite time and $A = -\frac{1}{2}$, i.e.

$$\mathbb{E}_{\text{KPZ}}\left[\exp(-\varsigma e^{H(t)})\right] = \mathbb{E}_{\text{GOE}}\left[\prod_{i=1}^{\infty} \frac{1}{\sqrt{1 + 4\varsigma e^{t^{1/3}a_i}}}\right], \tag{127}$$

where the set $\{a_i\}$ forms a Pfaffian GOE point process. The equivalence between the two formulae is under investigation.

# 6 Conclusion

In this paper we have extended the replica Bethe ansatz solution to the KPZ equation in a half-space for droplet initial condition near the wall, previously obtained for wall parameter $A = +\infty$ and $A = 0$, to generic value of $A$. This is an important problem which is believed, via universality, to exhibit a rich GSE/GOE/Gaussian transition first discovered by Baik and Rains. Despite much progress in the topic of exact solutions for the KPZ equation, this has remained a challenge for several years. A recent theorem which maps this problem to the (easier) case of $A = +\infty$ (Dirichlet) but with a Brownian initial condition was inspiring in the solution. Indeed we could apply the same type of summation as was developed in the solution of the Brownian initial condition in the full space. We used the known results for the Bethe ansatz eigenstates of the delta Bose gas in the half-line with Dirichlet boundary conditions.

At present, for a technical reason, our solution is valid for all times but limited to $A \geqslant -1/2$, i.e. the unbound phase. It allows to demonstrate the convergence to the GSE and GOE Tracy Widom distributions at large time in this phase. We have also obtained the crossover or transition kernel valid in the critical region $A + \frac{1}{2} = \epsilon t^{-1/3}$. The form of our transition kernel is novel, and we have shown that for $\epsilon \geqslant 0$ it agrees to a high order with the results of the kernel obtained by completely different methods for discrete models (facilitated TASEP and last passage percolation). This non-trivial check leads us to conjecture that these two kernel are fully equivalent. Additionally, let us note that the phase diagram we have presented for the solution to the KPZ equation echoes the one obtained in Ref. [41] Section 8 for the log-gamma polymer.

In a companion study [29] we have approached the problem differently by elucidating the structure of the boundary bound states which appear in the half-line Bose gas with generic parameter $A$. It quite easily yields results for the Gaussian phase $A < -1/2$ where these states dominate. It would be nice to find a bridge between these studies and to obtain a more complete solution of the KPZ equation in the half-space for all values of $A$. Let us close by mentioning that the physics content of the mapping between the Brownian initial condition and the droplet initial condition with different interactions with the wall remains to be understood deeper.

## Acknowledgements

We thank Guillaume Barraquand for countless discussions on the problem of the Kardar-Parisi-Zhang equation in a half-space. PLD acknowledges early discussions with A. Borodin, I. Corwin and P. Calabrese, and a fruitful ongoing collaboration on these and related topics with J. de Nardis and T. Thiery. We finally acknowledge support from ANR grant ANR-17-CE30-0027-01 RaMaTraF and AK acknowledges support from ERC under Consolidator grant number 771536 (NEMO).

## A   Overlap of the half-line Bethe states with the Brownian initial condition

Here we give some details on the calculation of the overlap $\langle \Psi_\mu | \Phi_0 \rangle$ between the half-line Bethe states and the Brownian initial condition. We recall that $\Phi_0$, given in (21), is a fully symmetric function of its arguments, and that in the sector $0 \leqslant x_1 < \cdots \leqslant x_n$ it equals

$$\Phi_0(x_1, \ldots, x_n) = \exp\left( \frac{1}{2} \sum_{j=1}^n (2n - 2j + 1)x_j - w x_j \right), \tag{128}$$

where $w = (A + \frac{1}{2})$ is the drift of the Brownian. Since we will find that the overlap is real, we will instead calculate its complex conjugate and use that $\langle \Psi_\mu | \Phi_0 \rangle^* = \langle \Phi_0 | \Psi_\mu \rangle = \langle \Psi_\mu | \Phi_0 \rangle$. Since $\Psi_\mu$ is also a symmetric function of its arguments, by definition the overlap can thus be written as

$$\langle \Phi_0 | \Psi_\mu \rangle = n! \int_{0 < y_1 < y_2 < \cdots < y_n} dy_1 \ldots dy_n \, \Psi_\mu(y_1, \ldots, y_n) e^{\sum_{j=1}^n \frac{1}{2}(2n+1-2j)y_j - w y_j}. \tag{129}$$

Inserting the explicit form of the Bethe eigenstate (23) as a superposition of plane waves we obtain

$$\langle \Phi_0 | \Psi_\mu \rangle = \frac{n!}{(2i)^n} \sum_{P \in S_n} \prod_{p=1}^n \left( \sum_{\varepsilon_p = \pm 1} \varepsilon_p \right) \tag{130}$$

$$\times \prod_{k < \ell} (1 + \frac{i}{\varepsilon_\ell \lambda_{P(\ell)} - \varepsilon_k \lambda_{P(k)}})(1 + \frac{i}{\varepsilon_\ell \lambda_{P(\ell)} + \varepsilon_k \lambda_{P(k)}}) G_{n,w}(\varepsilon_1 \lambda_{P(1)}, \ldots, \varepsilon_n \lambda_{P(n)}),$$

where we have defined the integrals

$$G_{n,w}(\lambda_1, \ldots, \lambda_n) = \int_{0 < y_1 < y_2 < \cdots < y_n} dy_1 \ldots dy_p \, e^{\sum_{j=1}^n (-w + i\lambda_j)y_j + \frac{1}{2}(2n+1-2j)y_j}. \tag{131}$$

These integrals can be explicitly evaluated

$$G_{n,w}(\lambda_1, \ldots, \lambda_n) = \prod_{j=1}^n \frac{-1}{-jw + i\lambda_n + \cdots + i\lambda_{n+1-j} + j^2/2}. \tag{132}$$

Now in (130) for each permutation $P$ we can relabel all the $\varepsilon_p \to \varepsilon_{P(p)}$ and denoting by $\sum_{\{\varepsilon\} = \pm 1}$ the operation of summation over all the variables $\varepsilon$ (an operation independent of

their labeling) we can rewrite (130) as

$$\langle \Phi_0 | \Psi_\mu \rangle = \frac{n!}{(2i)^n} \sum_{\{\varepsilon\}=\pm 1} \prod_{\ell=1}^n \varepsilon_\ell \prod_{k<\ell}(1 + \frac{i}{\varepsilon_k \lambda_k + \varepsilon_\ell \lambda_\ell}) \tag{133}$$

$$\sum_{P \in S_n} \prod_{k<\ell}(1 + \frac{i}{\varepsilon_{P(\ell)} \lambda_{P(\ell)} - \varepsilon_{P(k)} \lambda_{P(k)}}) G_{n,w}(\varepsilon_{P(1)} \lambda_{P(1)}, \ldots, \varepsilon_{P(n)} \lambda_{P(n)}),$$

where we have used the fact that the product

$$\prod_{k<\ell}(1 + \frac{i}{\varepsilon_{P(\ell)} \lambda_{P(\ell)} + \varepsilon_{P(k)} \lambda_{P(k)}}) = \prod_{k<\ell}(1 + \frac{i}{\varepsilon_k \lambda_k + \varepsilon_\ell \lambda_\ell}) \tag{134}$$

is independent of the permutation $P$. Now we use the following "miracle" equality, proved in [71]

$$\sum_{P \in S_n} \prod_{k<\ell}(1 + \frac{i}{\lambda_{P(\ell)} - \lambda_{P(n)}}) G_{p,w}(\lambda_{P(1)}, \ldots, \lambda_{P(n)}) = \prod_{j=1}^n \frac{1}{w - \frac{1}{2} - i\lambda_j}. \tag{135}$$

Applying it to the set $\{\varepsilon_k \lambda_k\}$ we obtain

$$\langle \Phi_0 | \Psi_\mu \rangle = \frac{n!}{(2i)^n} \sum_{\{\varepsilon\}=\pm 1} \prod_{k<\ell}(1 + \frac{i}{\varepsilon_k \lambda_k + \varepsilon_\ell \lambda_\ell}) \prod_{j=1}^n \frac{\varepsilon_j}{w - \frac{1}{2} - i\varepsilon_j \lambda_j}. \tag{136}$$

Let us denote $w = A + \frac{1}{2}$. We now conjecture the following identity valid for any set of complex numbers $\lambda_j$

$$\sum_{\{\varepsilon\}=\pm 1} \prod_{k<\ell}(1 + \frac{i}{\varepsilon_k \lambda_k + \varepsilon_\ell \lambda_\ell}) \prod_{j=1}^n \frac{\varepsilon_j}{A - i\varepsilon_j \lambda_j} = (2i)^n \prod_{j=1}^n \frac{\lambda_j}{A^2 + \lambda_j^2}, \tag{137}$$

which we have checked to high order $n = 5$ using Mathematica. Inserting in (136) we obtain the formula for the overlap (34) given in the text.

We note that the overlap formula it is a priori valid before the insertion of the solution of the Bethe equations, i.e. it is valid for any set of complex $\lambda_j$, invariant by complex conjugation, such that the overlap integral converges. The condition for that to be true can be read from (135) as

$$\text{Re}(i\lambda_j) < w - \frac{1}{2} \tag{138}$$

for all $j \in [1, n]$. Inserting a string state, labeled by $\{k_j, m_j\}_{j=1,\ldots,n_s}$, and specifying $w = A + \frac{1}{2}$, the condition becomes $\frac{1}{2} \max_j m_j \leqslant \frac{n}{2} < A + \frac{1}{2}$.

## B  Calculation of the first moment of $Z(t)$

Starting from the general formula (38) for the moments of $Z(t)$ we can read the explicit expression for $\mathbb{E}_{\text{KPZ}}[Z(t)]$, i.e. $n = 1$, for $A > 0$.

$$\mathbb{E}_{\text{KPZ}}[Z(t)]|_{(38)} = \int_{\mathbb{R}} \frac{dk}{\pi} e^{-k^2 t} \frac{k^2}{A^2 + k^2}. \tag{139}$$

This result however is not correct for $A < 0$ as we now show.

The Lieb-Liniger model for a single particle reduces to the standard heat equation on the half-line with boundary coefficient $A$, which is easily solved. The (un-normalized) eigenfunctions are the

- For all $A \in \mathbb{R}$, the following plane wave solves the eigenvalue equation with eigenenergy $k^2$

$$\Psi(x) = \frac{1}{2i}\left(e^{ikx}(1+i\frac{k}{A}) - e^{-ikx}(1-i\frac{k}{A})\right). \tag{140}$$

Its norm on a line of length $L$ is $||\Psi||^2 = \frac{L}{2}(1+\frac{k^2}{A^2})$. The real momentum $k$ is quantized on the line $[0, L]$. In the $L \to \infty$ limit, the summation over eigenstates becomes $\sum_k \to L \int_{\mathbb{R}} \frac{dk}{2\pi}$. The contribution of this plane wave to the first moment of $Z(t)$ is

$$Z_A(t) = L \int_{\mathbb{R}} \frac{dk}{2\pi} e^{-k^2 t} \frac{\Psi(0)^2}{||\Psi||^2} = \int_{\mathbb{R}} \frac{dk}{\pi} e^{-k^2 t} \frac{k^2}{A^2 + k^2}. \tag{141}$$

- For all $A < 0$, an additional state, i.e. a boundary bound state (BS), with energy $-A^2$ appears

$$\Psi_{BS}(x) = e^{Ax}. \tag{142}$$

Its norm on the infinite line is $||\Psi_{BS}||^2 = -\frac{1}{2A}$ and its contribution to the first moment of $Z(t)$ is

$$Z_{BS}(t) = e^{A^2 t} \frac{\Psi_{BS}(0)^2}{||\Psi_{BS}||^2} = -2A e^{A^2 t} \Theta(-A), \tag{143}$$

where $\Theta$ is the Heaviside function.

The total result valid for all $A \in \mathbb{R}$ is thus

$$\mathbb{E}_{\text{KPZ}}\big[Z(t)\big] = \int_{\mathbb{R}} \frac{dk}{\pi} e^{-k^2 t} \frac{k^2}{A^2 + k^2} - 2A e^{A^2 t} \Theta(-A). \tag{144}$$

This can also be obtained by solving the heat equation in time, leading to the same result.

We observe that the second contribution $Z_{BS}$ is absent from our result (38). It confirms that our moment formula is valid only in a certain range of values, i.e. $A + 1/2 > n/2$ for the $n$-th moment of $Z(t)$. The complete formula thus requires including the boundary bound states.

# C  Comparison of the Fredholm Pfaffians

As announced in the main text, we now test our conjecture (74), i.e. we compare the Fredholm Pfaffian based on our transition kernel, given in (72), and the one based on the crossover kernel of Ref. [37] Definition 2.9, recalled in (75).

## C.1  Series expansion of a Fredholm Pfaffian

Let us recall here the first terms in the series expansion of a Fredholm Pfaffian in powers of its kernel. Here $K$ denotes a generic kernel. By definition

$$\text{Pf}(J - K)_{\mathbb{L}^2([s,+\infty[)} = 1 + \sum_{n_s=1}^{\infty} \frac{(-1)^{n_s}}{n_s!} \prod_{p=1}^{n_s} \int_s^{+\infty} dr_p \, \text{Pf}[K(r_i, r_j)]_{n_s, n_s}. \tag{145}$$

More explicitly we write (with the convention that $K_{21}(r,r') = -K_{12}(r',r)$)

$$\mathrm{Pf}[J-K]_{\mathbb{L}^2([s,+\infty[)} = 1 - \int_s^{+\infty} dr\, \mathrm{Pf}\begin{pmatrix} 0 & K_{12}(r,r) \\ -K_{12}(r,r) & 0 \end{pmatrix}$$

$$+ \frac{1}{2!} \iint_s^{+\infty} dr_1 dr_2\, \mathrm{Pf}\begin{pmatrix} 0 & K_{12}(r_1,r_1) & K_{11}(r_1,r_2) & K_{12}(r_1,r_2) \\ K_{21}(r_1,r_1) & 0 & K_{21}(r_1,r_2) & K_{22}(r_1,r_2) \\ K_{11}(r_2,r_1) & K_{12}(r_2,r_1) & 0 & K_{12}(r_2,r_2) \\ K_{21}(r_2,r_1) & K_{22}(r_2,r_1) & K_{21}(r_2,r_2) & 0 \end{pmatrix}$$

$$- \frac{1}{3!} \iiint_s^{+\infty} dr_1 dr_2 dr_3$$

$$\mathrm{Pf}\begin{pmatrix} 0 & K_{12}(r_1,r_1) & K_{11}(r_1,r_2) & K_{12}(r_1,r_2) & K_{11}(r_1,r_3) & K_{12}(r_1,r_3) \\ K_{21}(r_1,r_1) & 0 & K_{21}(r_1,r_2) & K_{22}(r_1,r_2) & K_{21}(r_1,r_3) & K_{22}(r_1,r_3) \\ K_{11}(r_2,r_1) & K_{12}(r_2,r_1) & 0 & K_{12}(r_2,r_2) & K_{11}(r_2,r_3) & K_{12}(r_2,r_3) \\ K_{21}(r_2,r_1) & K_{22}(r_2,r_1) & K_{21}(r_2,r_2) & 0 & K_{21}(r_2,r_3) & K_{22}(r_2,r_3) \\ K_{11}(r_3,r_1) & K_{12}(r_3,r_1) & K_{11}(r_3,r_2) & K_{12}(r_3,r_2) & 0 & K_{12}(r_3,r_3) \\ K_{21}(r_3,r_1) & K_{22}(r_3,r_1) & K_{21}(r_3,r_2) & K_{22}(r_3,r_2) & K_{21}(r_3,r_3) & 0 \end{pmatrix} + \mathcal{O}(K^4)$$

$$= 1 - \mathrm{Tr}K_{12} + \frac{1}{2}\left[ (\mathrm{Tr}K_{12})^2 - \mathrm{Tr}K_{12}^2 + \mathrm{Tr}K_{11}K_{22} \right]$$

$$- \frac{1}{6}\left[ (\mathrm{Tr}K_{12})^3 + 2\mathrm{Tr}K_{12}^3 - 3\mathrm{Tr}K_{12}\mathrm{Tr}K_{12}^2 + 3\mathrm{Tr}K_{12}\mathrm{Tr}K_{11}K_{22} - 6\mathrm{Tr}K_{12}K_{11}K_{22} \right] + \mathcal{O}(K^4),$$

$$(146)$$

where all integrations in the traces run onto $[s,+\infty[$. For completeness, we also provide the fourth order of the expansion, as given in Appendix G of Ref. [13].

$$\mathcal{O}(K^4) = \frac{1}{4!}\Big( \mathrm{Tr}[K_{12}]^4 - 6\mathrm{Tr}[K_{12}]^2\mathrm{Tr}[K_{12}^2] + 3\mathrm{Tr}[K_{12}^2]^2 + 8\mathrm{Tr}[K_{12}]\mathrm{Tr}[K_{12}^3]$$

$$- 6\mathrm{Tr}[K_{12}^4] + 6\mathrm{Tr}[K_{12}]^2\mathrm{Tr}[K_{11}K_{22}] - 6\mathrm{Tr}[K_{12}^2]\mathrm{Tr}[K_{11}K_{22}] + 3\mathrm{Tr}[K_{11}K_{22}]^2$$

$$- 6\mathrm{Tr}[(K_{11}K_{22})^2] - 24\mathrm{Tr}[K_{12}]\mathrm{Tr}[K_{11}K_{22}K_{12}] + 24\mathrm{Tr}[K_{11}K_{22}K_{12}^2]$$

$$+ 12\mathrm{Tr}[K_{11}K_{12}^{\mathrm{T}}K_{22}K_{12}] \Big).$$

$$(147)$$

*Remark* C.1. In the case where $K_{12}$ is rank 1 operator, we have for all integer $n$, $\mathrm{Tr}(K_{12}^n) = (\mathrm{Tr}K_{12})^n$, simplifying the expansion dramatically. In that case it becomes [13]

$$\mathrm{Pf}[J-K]_{\mathbb{L}^2([s,+\infty[)} = 1 - \mathrm{Tr}K_{12} + \frac{1}{2}\mathrm{Tr}K_{11}K_{22} + \frac{1}{2}(2\mathrm{Tr}K_{12}K_{11}K_{22} - \mathrm{Tr}K_{12}\mathrm{Tr}K_{11}K_{22})$$

$$+ \frac{1}{4!}(3(\mathrm{Tr}K_{11}K_{22})^2 - 6\mathrm{Tr}(K_{11}K_{22})^2) + \mathcal{O}(K^5).$$

$$(148)$$

## C.2 Kernel with all contours at the right of $\epsilon$

To be able to track systematically the degree of homogeneity of each integral in the expansion at large $s$ as explained in the text, we move all integration contours to $\hat{C}$ which is at the right of $\epsilon$. For our kernel this was done in (72), leading to the schematic form (73) where the upper label $^{(p)}$, $p = 1,2$ denotes the number of independent integrals with a cubic exponential, leading to the large positive $s$ decay $\sim \exp(\frac{2}{3}ps^{3/2})$. We need to put the kernel $K_{\mathrm{cross}}$ in (75)

in the same form. It reads

$$
\begin{aligned}
I_{11}(r,r') &= \iint_{\hat{C}^2} \frac{\mathrm{d}z}{2i\pi} \frac{\mathrm{d}w}{2i\pi} \frac{w-z}{w+z} \frac{w+\epsilon}{w} \frac{z+\epsilon}{z} e^{-r'z-rw+\frac{w^3+z^3}{3}} \\
I_{12}(r,r') &= \frac{1}{2} \iint_{\hat{C}^2} \frac{\mathrm{d}z}{2i\pi} \frac{\mathrm{d}w}{2i\pi} \frac{w-z}{w+z} \frac{w+\epsilon}{w} \frac{1}{\epsilon-z} e^{-r'z-rw+\frac{w^3+z^3}{3}} \\
&\qquad + \frac{1}{2} \int_{\hat{C}} \frac{\mathrm{d}w}{2i\pi} \frac{w-\epsilon}{w} e^{-r'\epsilon-rw+\frac{w^3+\epsilon^3}{3}} \\
I_{22}(r,r') &= \frac{1}{4} \iint_{\hat{C}^2} \frac{\mathrm{d}z}{2i\pi} \frac{\mathrm{d}w}{2i\pi} \frac{w-z}{w+z} \frac{1}{\epsilon-z} \frac{1}{\epsilon-w} e^{-r'z-rw+\frac{w^3+z^3}{3}} \\
&\qquad + \frac{1}{4} \int_{\hat{C}} \frac{\mathrm{d}z}{2i\pi} \frac{e^{z^3/3+\epsilon^3/3}}{\epsilon+z} \left[ e^{-r'z-r\epsilon} - e^{-rz-r'\epsilon} \right] \\
R_{22}(r,r') &= \frac{1}{4} \mathrm{sgn}(r'-r) e^{-|r-r'|\epsilon},
\end{aligned}
\tag{149}
$$

where, as mentionned in the text the total $K_{\mathrm{cross},22}$ element of $K_{\mathrm{cross}}$ is equal to the sum $I_{22}+R_{22}$, hence for convenience we have moved some terms from $R_{22}$ to $I_{22}$. The decomposition in their homogeneity in cubic exponentials (without counting the terms $\epsilon^3$) read

$$
\begin{aligned}
I_{11} &= I_{11}^{(2)} \\
I_{22} + R_{22} &= I_{22}^{(2)} + I_{22}^{(1)} + R_{22}^{(0)} \\
I_{12} &= I_{12}^{(2)} + I_{12}^{(1)}.
\end{aligned}
\tag{150}
$$

The important observation is that there is an anomalous term, of homogeneity zero $p = 0$, coming from $R_{22}$.

*Remark* C.2. $I_{12}^{(1)}$ is a rank 1 operator.

## C.3 Expansion of the Pfaffians in degree of homogeneity

Consider first the expansion of $\mathrm{Pf}(J - K^\epsilon)_{\mathbb{L}^2([s,+\infty[)}$ using the formula (146) (and if needed (160) and (148)). We expand in the total degree of homogeneity (i.e. the degree of stretched exponential decay at large $s$) so we need to pick up terms in various orders in the expansion of the Pfaffian. Fortunately, to expand to third order in homogeneity this order is not too large. Recalling that

$$
\begin{aligned}
K_{11}^\epsilon &= K_{11}^{\epsilon,(2)} + K_{11}^{\epsilon,(1)} \\
K_{22}^\epsilon &= K_{22}^{\epsilon,(2)} \\
K_{12}^\epsilon &= K_{12}^{\epsilon,(2)} + K_{12}^{\epsilon,(1)}
\end{aligned}
\tag{151}
$$

and gathering terms of successive orders of homogeneity in (146), we obtain (here $K$ denotes $K^\epsilon$)

$$
\begin{aligned}
\text{order } \mathcal{O}(0): &\quad 1 \\
\text{order } \mathcal{O}(1): &\quad -\mathrm{Tr}K_{12}^{(1)} \\
\text{order } \mathcal{O}(2): &\quad -\mathrm{Tr}K_{12}^{(2)} \\
\text{order } \mathcal{O}(3): &\quad \frac{1}{2}\mathrm{Tr}K_{11}^{(1)}K_{22}^{(2)} + \mathrm{Tr}K_{12}^{(2)}\mathrm{Tr}K_{12}^{(1)} - \mathrm{Tr}K_{12}^{(2)}K_{12}^{(1)}.
\end{aligned}
\tag{152}
$$

Performing the same operation for $\text{Pf}(J - K_{\text{cross}})_{\mathbb{L}^2([s,+\infty[)}$, inserting (150) into (146) we obtain

$$
\begin{aligned}
&\text{order } \mathcal{O}(0): && 1 \\
&\text{order } \mathcal{O}(1): && -\text{Tr}I_{12}^{(1)} \\
&\text{order } \mathcal{O}(2): && -\text{Tr}I_{12}^{(2)} + \frac{1}{2}\text{Tr}R_{22}^{(0)}I_{11}^{(2)} \\
&\text{order } \mathcal{O}(3): && \frac{1}{2}\text{Tr}I_{11}^{(2)}I_{22}^{(1)} + \text{Tr}I_{12}^{(2)}\text{Tr}I_{12}^{(1)} - \text{Tr}I_{12}^{(2)}I_{12}^{(1)} \\
& && \qquad - \frac{1}{2}\left[\text{Tr}I_{12}^{(1)}\text{Tr}I_{11}^{(2)}R_{22}^{(0)} - 2\text{Tr}I_{12}^{(1)}I_{11}^{(2)}R_{22}^{(0)}\right].
\end{aligned}
\tag{153}
$$

Thus, to verify that the two Fredholm Pfaffians in (74) are equal to third order, i.e. to and including $\mathcal{O}(e^{-2s^{3/2}})$, we need to show the following equalities

$$
\begin{aligned}
&\text{Tr}K_{12}^{(1)} = \text{Tr}I_{12}^{(1)} \\
&\text{Tr}K_{12}^{(2)} = \text{Tr}I_{12}^{(2)} - \frac{1}{2}\text{Tr}R_{22}^{(0)}I_{11}^{(2)} \\
&\frac{1}{2}\text{Tr}K_{11}^{(1)}K_{22}^{(2)} + \text{Tr}K_{12}^{(2)}\text{Tr}K_{12}^{(1)} - \text{Tr}K_{12}^{(2)}K_{12}^{(1)} \\
&= \frac{1}{2}\text{Tr}I_{11}^{(2)}I_{22}^{(1)} + \text{Tr}I_{12}^{(2)}\text{Tr}I_{12}^{(1)} - \text{Tr}I_{12}^{(2)}I_{12}^{(1)} - \frac{1}{2}\left[\text{Tr}I_{12}^{(1)}\text{Tr}I_{11}^{(2)}R_{22}^{(0)} - 2\text{Tr}I_{12}^{(1)}I_{11}^{(2)}R_{22}^{(0)}\right].
\end{aligned}
\tag{154}
$$

Below we will prove the derivatives w.r.t. $s$ of these equalities, which is simpler in general and sufficient since all terms vanish at $s = +\infty$. Once we have shown the first two equalities, we can use them to simplify the third one, so we will need to show

$$
\frac{1}{2}\text{Tr}K_{11}^{(1)}K_{22}^{(2)} - \text{Tr}K_{12}^{(2)}K_{12}^{(1)} = \frac{1}{2}\text{Tr}I_{11}^{(2)}I_{22}^{(1)} - \text{Tr}I_{12}^{(2)}I_{12}^{(1)} + \text{Tr}I_{12}^{(1)}I_{11}^{(2)}R_{22}^{(0)}.
\tag{155}
$$

### C.3.1 Comparison of terms of homogeneity 1

One sees from (72) and (149) that $I_{12}^{(1)}(r, r') = K_{12}^{(1)}(r', r)$ hence one has immediately

$$
\text{Tr}I_{12}^{(1)} = \text{Tr}K_{12}^{(1)}.
\tag{156}
$$

The first equality in (154) then holds.

### C.3.2 Comparison of terms of homogeneity 2

From (154) we now have to compare

$$
\int_s^\infty dr\, K_{12}^{(2)}(r, r) \stackrel{?}{=} \int_s^\infty dr\, I_{12}^{(2)}(r, r) - \frac{1}{2}\iint_s^\infty dr\, dr'\, I_{11}^{(2)}(r, r')R_{22}^{(0)}(r', r).
\tag{157}
$$

Taking the derivative with respect to $s$, and using the anti-symmetry of $I_{11}$ and $R_{22}$ this is equivalent to

$$
K_{12}^{(2)}(s, s) \stackrel{?}{=} I_{12}^{(2)}(s, s) - \int_s^\infty dr'\, I_{11}^{(2)}(s, r')R_{22}^{(0)}(r', s).
\tag{158}
$$

We recall that

$$
K_{12}^{(2)}(r, r') = \frac{1}{4}\iint_{\hat{C}^2} \frac{dw}{2i\pi}\frac{dz}{2i\pi}\frac{w - z}{w + z}\frac{z + \epsilon}{wz}\frac{\epsilon + w}{w - \epsilon}e^{-rw - r'z + \frac{w^3 + z^3}{3}}
\tag{159}
$$

and

$$I_{12}^{(2)}(r,r') = \frac{1}{2}\iint_{\hat{C}^2} \frac{dz}{2i\pi}\frac{dw}{2i\pi}\frac{w-z}{w+z}\frac{w+\epsilon}{w}\frac{1}{\epsilon-z}e^{-r'z-rw+\frac{w^3+z^3}{3}}$$ (160)

and

$$I_{11}^{(2)}(r,r') = \iint_{\hat{C}^2} \frac{dz}{2i\pi}\frac{dw}{2i\pi}\frac{w-z}{w+z}\frac{w+\epsilon}{w}\frac{z+\epsilon}{z}e^{-r'z-rw+\frac{w^3+z^3}{3}}$$ (161)

and

$$R_{22}^{(0)}(r,r') = \frac{1}{4}\text{sgn}(r'-r)e^{-|r-r'|\epsilon}.$$ (162)

Since the integrals are on $[s,+\infty[$ one has $R_{22}^{(0)}(r',s) = -\frac{1}{4}e^{-(r'-s)\epsilon}$. Evaluating the integral we obtain

$$\int_s^\infty dr' I_{11}^{(2)}(s,r')R_{22}^{(0)}(r',s) = -\frac{1}{4}\int_s^\infty dr' \iint_{\hat{C}^2}\frac{dz}{2i\pi}\frac{dw}{2i\pi}\frac{w-z}{w+z}\frac{w+\epsilon}{w}\frac{z+\epsilon}{z}e^{-r'(z+\epsilon)-sw+s\epsilon+\frac{w^3+z^3}{3}}$$
$$= -\frac{1}{4}\iint_{\hat{C}^2}\frac{dz}{2i\pi}\frac{dw}{2i\pi}\frac{w-z}{w+z}\frac{w+\epsilon}{wz}e^{-sz-sw+\frac{w^3+z^3}{3}}.$$ (163)

Inserting into (158) and using the expressions (159), (160) we see that the measure in the integrals including the exponentials are symmetric in $(w,z)$. Hence the equality (158) will hold if we can show that the preexponential factor once symmetrized over $(w,z)$ vanishes. The condition reads

$$\text{sym}_{w,z}\frac{w-z}{w+z}\frac{\epsilon+w}{w}\left(\frac{1}{4}\frac{z+\epsilon}{z}\frac{1}{w-\epsilon} - \frac{1}{2}\frac{1}{\epsilon-z} - \frac{1}{4z}\right) = 0$$ (164)

and is easily checked to be true using mathematica. Hence the second equality in (154) holds.

### C.3.3 Comparison of terms of homogeneity 3

We now calculate each term appearing in (155). One has

$$\text{TrK}_{12}^{(2)}K_{12}^{(1)}$$
$$= \frac{1}{8}\iint_s^{+\infty}drdr'\iiint_{\hat{C}^3}\frac{dv}{2i\pi}\frac{dw}{2i\pi}\frac{dz}{2i\pi}\frac{w-z}{w+z}\frac{z+\epsilon}{wz}\frac{\epsilon+w}{w-\epsilon}\frac{v-\epsilon}{v}e^{-r(w+v)-r'(z+\epsilon)+\frac{w^3+z^3+v^3+\epsilon^3}{3}}$$
$$= \frac{1}{8}\iiint_{\hat{C}^3}\frac{dv}{2i\pi}\frac{dw}{2i\pi}\frac{dz}{2i\pi}\frac{w-z}{w+z}\frac{1}{wvz}\frac{\epsilon+w}{w-\epsilon}\frac{v-\epsilon}{v+w}e^{-s(w+v+z+\epsilon)+\frac{w^3+z^3+v^3+\epsilon^3}{3}}$$ (165)

and

$$\text{TrK}_{11}^{(1)}K_{22}^{(2)}$$
$$= \frac{1}{8}\iint_s^{+\infty}drdr'\iiint_{\hat{C}^3}\frac{dv}{2i\pi}\frac{dw}{2i\pi}\frac{dz}{2i\pi}\left[e^{-rv-r'\epsilon} - e^{-r'v-r\epsilon}\right]$$
$$\times\frac{w-z}{w+z}\frac{(z+\epsilon)(w+\epsilon)}{wvz}e^{-rw-r'z+\frac{w^3+z^3+v^3+\epsilon^3}{3}}$$ (166)
$$= \frac{1}{8}\iiint_{\hat{C}^3}\frac{dv}{2i\pi}\frac{dw}{2i\pi}\frac{dz}{2i\pi}\left[\frac{1}{(v+w)(z+\epsilon)} - \frac{1}{(z+v)(w+\epsilon)}\right]$$
$$\times\frac{w-z}{w+z}\frac{(z+\epsilon)(w+\epsilon)}{wvz}e^{-s(w+z+v+\epsilon)+\frac{w^3+z^3+v^3+\epsilon^3}{3}}$$

and

$$
\mathrm{TrI}^{(2)}_{12}\mathrm{I}^{(1)}_{12}
$$
$$
= \frac{1}{4}\iint_s^{+\infty} \mathrm{d}r\mathrm{d}r' \iiint_{\hat{C}^3} \frac{\mathrm{d}z}{2i\pi}\frac{\mathrm{d}w}{2i\pi}\frac{\mathrm{d}v}{2i\pi}\frac{w-z}{w+z}\frac{w+\epsilon}{w}\frac{1}{\epsilon-z}\frac{v-\epsilon}{v}e^{-r'(z+v)-r(w+\epsilon)+\frac{w^3+z^3+v^3+\epsilon^3}{3}}
$$
$$
= \frac{1}{4}\iiint_{\hat{C}^3} \frac{\mathrm{d}z}{2i\pi}\frac{\mathrm{d}w}{2i\pi}\frac{\mathrm{d}v}{2i\pi}\frac{w-z}{w+z}\frac{1}{wv}\frac{1}{\epsilon-z}\frac{v-\epsilon}{v+z}e^{-s(z+v+w+\epsilon)+\frac{w^3+z^3+v^3+\epsilon^3}{3}}
$$

(167)

and

$$
\mathrm{TrI}^{(2)}_{11}\mathrm{I}^{(1)}_{22}
$$
$$
= \frac{1}{4}\iint_s^{+\infty} \mathrm{d}r\mathrm{d}r' \iiint_{\hat{C}^3} \frac{\mathrm{d}z}{2i\pi}\frac{\mathrm{d}w}{2i\pi}\frac{\mathrm{d}v}{2i\pi}\frac{1}{\epsilon+v}\frac{w-z}{w+z}\frac{w+\epsilon}{w}\frac{z+\epsilon}{z}\left[e^{-rv-r'\epsilon}-e^{-r'v-r\epsilon}\right]
$$
$$
\times e^{-r'z-rw+\frac{w^3+z^3+v^3+\epsilon^3}{3}}
$$

(168)

$$
= \frac{1}{4}\iiint_{\hat{C}^3} \frac{\mathrm{d}z}{2i\pi}\frac{\mathrm{d}w}{2i\pi}\frac{\mathrm{d}v}{2i\pi}\frac{1}{\epsilon+v}\frac{w-z}{w+z}\frac{w+\epsilon}{w}\frac{z+\epsilon}{z}\left[\frac{1}{(v+w)(\epsilon+z)}-\frac{1}{(v+z)(w+\epsilon)}\right]
$$
$$
\times e^{-s(v+w+z+\epsilon)+\frac{w^3+z^3+v^3+\epsilon^3}{3}}
$$

and, finally

$$
\mathrm{TrI}^{(1)}_{12}\mathrm{I}^{(2)}_{11}\mathrm{R}^{(0)}_{22}
$$
$$
= \frac{1}{8}\iiint_s^{+\infty} \mathrm{d}r\mathrm{d}r'\mathrm{d}r'' \iiint_{\hat{C}^3} \frac{\mathrm{d}z}{2i\pi}\frac{\mathrm{d}w}{2i\pi}\frac{\mathrm{d}v}{2i\pi}\frac{w-z}{w+z}\frac{w+\epsilon}{w}\frac{z+\epsilon}{z}\frac{v-\epsilon}{v}\mathrm{sgn}(r''-r')
$$
$$
\times e^{-|r'-r''|\epsilon-r(\epsilon+w)-r''v-r'z+\frac{w^3+z^3+v^3+\epsilon^3}{3}}
$$

(169)

$$
= \frac{1}{8}\iint_s^{+\infty} \mathrm{d}r'\mathrm{d}r'' \iiint_{\hat{C}^3} \frac{\mathrm{d}z}{2i\pi}\frac{\mathrm{d}w}{2i\pi}\frac{\mathrm{d}v}{2i\pi}\frac{w-z}{w+z}\frac{(z+\epsilon)(v-\epsilon)}{vwz}\mathrm{sgn}(r''-r')
$$
$$
\times e^{-|r'-r''|\epsilon-s(\epsilon+w)-r''v-r'z+\frac{w^3+z^3+v^3+\epsilon^3}{3}}
$$

$$
= \frac{1}{8}\iiint_{\hat{C}^3} \frac{\mathrm{d}z}{2i\pi}\frac{\mathrm{d}w}{2i\pi}\frac{\mathrm{d}v}{2i\pi}\frac{w-z}{w+z}\frac{z-v}{v+z}\frac{v-\epsilon}{vwz}\frac{1}{v+\epsilon}e^{-s(\epsilon+z+v+w)+\frac{w^3+z^3+v^3+\epsilon^3}{3}}.
$$

Putting now all terms together as in (155) and symmetrizing the preexponential factor in $(w,z)$ as we did for the terms of homogeneity 2, we find, after a tedious calculation (using mathematica) that the symmetrized form vanishes. We believe that the agreement of the first three orders is a rather non-trivial check, which comes in very strong support of our conjecture (74).

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
