# Peer review of "Replica Bethe Ansatz solution to the Kardar-Parisi-Zhang equation on the half-line"

_SciPost Physics, doi:SciPost Phys. 8, 035 (2020)_

## Round 3 · Referee Report · Anonymous · 2019-11-29

Strengths

1. Leading edge results of great importance
2. Very High technical level
3. Extremely well written

Report

The authors study the KPZ equation on the positive half-line with
a Neumann boundary condition at $x=0$, $\partial_x h(0,t) = A$,
and droplet initial condition. Inspired by a very recent theorem of Parekh, the authors solve the equivalent problem of KPZ on the half-line with Dirichlet boundary condition at $x=0$ but in which the initial condition is a Brownian motion with drift $A + 1/2$. A full solution is obtained for positive drift. The central result is an exact formula for the Laplace transform of the distribution of the exponential of the height at 0, valid at all times.This result is expressed as a (Fredholm) Pfaffian of a matrix kernel. The authors also show (in section 5) that their formula can also be written as the square-root of a (Fredholm) Determinant of a scalar kernel.
The large time asymptotics is also investigated : the generic case leads to
the GSE TW distribution; the limiting case $A + 1/2 =0$ gives the GOE.
The crossover is also studied in details, leading to a new expression
for the cross-over kernel $K^{\epsilon}$ which is conjectured to be equivalent to another kernel $K_{\rm{cross}}$ that has appeared in various recent works.

The strategy used by the authors is based on the replica Bethe Ansatz, a technique one that one the authors -- with different collaborators - successfully applied a few years ago to solve the KPZ equation (simultaneously with other groups). The main difficulty being to determine the overlap of the Bethe states with the initial condition. An exact formula is obtained in Appendix A, based on a generalization of a ``magic formula'' derived by Imamura and Sasamoto.

This paper presents leading edge results and is extremely well written. The logical line followed by the authors is very clearly stated. The authors are very explicit and very honest about their assumptions and the (present) limitations of their results. All important details are given and, with some patience, it is possible to understand and to reproduce the calculations of the manuscript. I have no specific remarks apart from:

(i) a possible typo in eq. (23): I do not understand why $A$ depends on the $\epsilon_i$'s.

(ii) It took me time to understand that in eq. (28) what is written is
a product of two fractions (and not three terms in the numerator
and three terms in the denominator). I had to go to ref [24].
May be you could add some space between the two fractions.

(iii) In the exponential in eq. (29): $ m_j \rightarrow m_p$.

(iv) May be it could be useful to add a few words to explain the
continuum limit $\sum_{k_j} \rightarrow m_j \int_{\mathbb{ R}} \frac{d k}{2 \pi}$.

(v) A crucial step is given in eqs (39) and (40). However, no reference
and no proof of the Schur Pfaffian are given. I checked
it with Mathematica for small $n_s$ but did not find a proof. It could
be nice to have some clue here.

(vi) The authors use the Mellin-Barnes summation formula before eq. (45).
It would be useful to explain where and how they overcome the barrier
$A >(n-1)/2$.

I wholeheartedly recommend this manuscript for publication.
This is an impressive work, beautifully presented.

Attachment

---

## Round 3 · Referee Report · Anonymous · 2019-12-8

Strengths

1. A solid progress on an important but hard problem

2. Well written

Report

In this paper the authors study the KPZ equation on the half-line using the replica Bethe ansatz.
Compared to the KPZ equation on the whole line, studies for the model on the half-line has been restricted to a few particular special cases. In this paper the authors succeeded in generalizing
such results to the case with the Neumann boundary condition for much broader range of the
boundary parameter.

Their main results is the exact Fredholm Pfaffian expression for the Laplace transform of the height
function. With this formula, the authors discuss various aspects of the problem such as the long time limits, specializations to previously known cases and the transition regions. The paper is well written. The motivations are clearly explained, previous works are explained well, the deviations are detailed enough, and many interesting issues related to the problem are addressed.

The reviewer could recommend a publication of the article as it is (after taking into account the comments by the other referee).

---

## Round 4 · Author Response

Dear Editor and Referees,

We are grateful for the efforts in reviewing our manuscript. We thank the referees for their constructive comments of our paper. In the following, we believe we answer the concerns raised by the first referee and list all changes we made in the resubmitted version.

Please let us know if any other information could be helpful.

Sincerely yours,
Alexandre Krajenbrink and Pierre Le Doussal

---

## Round 4 · List of Changes

Warnings issued while processing user-supplied markup:

  • Inconsistency: plain/Markdown and reStructuredText syntaxes are mixed. Markdown will be used.
    Add "#coerce:reST" or "#coerce:plain" as the first line of your text to force reStructuredText or no markup.
    You may also contact the helpdesk if the formatting is incorrect and you are unable to edit your text.

Response to Referee 1

The referee raised several remarks to which we answer:

(i) The dependence of the amplitude A on the parity of the rapidities $\lambda_j$ is not a typo. Each rapidity comes in pair with its opposite since we study a hard-wall geometry: the image method imposes to sum over the parity of the rapidity. Nonetheless we agree that the parentheses are confusing in Eq. (23) hence it has been modified for clarity.

(ii) We agree that the fraction requires some more space so we added parenthesis and an extra multiplication sign for clarity.

(iii) The typo has been changed, we that the referee for pointing this out.

(iv) We agree with the referee that a few words could be added to explain the continuum limit and hence we added the sentence « the momenta sums become continuous and one can use that the string momenta $m_j k_j$ correspond to free particles.» and quoted a number papers where the exact same replacement was made.

(v) We have added a classical reference from Knuth to Pfaffians which includes the Schur Pfaffian and have transformed the equation for the Schur Pfaffian from an inline equation to a numbered equation. The main remark we have concerning this Pfaffian is that for the KPZ equation in full-space, the inverse of the norm of the Bethe states are given by a Cauchy determinant whereas in the half-space it is given by a Schur Pfaffian which allows to obtain an explicit solution of the generating function of the exponential of the KPZ height in terms of a Fredholm Pfaffian.

(vi) The Mellin-Barnes summation applied in our work is inspired by the one carried out by Sasamoto and Imamura in the study of the stationary initial condition for the KPZ equation in full-space. In their case, a similar requirement on the drift of the Brownian initial condition was expressed and was shown to be lifted after the application of a Mellin-Barnes summation in conjunction with an analytic continuation of Gamma functions and the right choice of contour in the complex plane. We entirely agree with the referee that this passage deserves some more details and hence we have added a precise reference to the work of Sasamoto and Imamura to find the original application of Mellin-Barnes for this case and we have added the sentence: « The summation over the variables $m_p$ can be done using the Mellin-Barnes summation trick similarly to Refs. [paper of Sasamoto and Imamura]. The barrier $A>(n-1)/2$ is overcome exactly as in Ref.~[paper of Sasamoto and Imamura] (see Lemma.~6 and the discussion following there) from an analytic continuation of Gamma functions in the $B_{k,m}$ factor, the introduction of a particular contour $C_0$ and a final requirement for the drift $A+1/2>0$. »

---

## Editorial Decision

published